# *Entamoeba histolytica* extracellular vesicles drive pro-inflammatory monocyte signaling

**Barbara Honecker** [1,2*], **Valentin A. Bärreiter** [1,3,4,5], **Katharina Höhn** [6], **Balázs Horváth** [7], **Karel Harant** [8], **Nahla Galal Metwally** [2], **Claudia Marggraff** [1], **Juliett Anders** [2], **Stephanie Leyk** [9,10], **Maria del Pilar Martínez-Tauler** [2,11], **Annika Bea** [1], **Charlotte Hansen** [1], **Helena Fehling** [1], **Melanie Lütkemeyer** [1], **Stephan Lorenzen** [12], **Sören Franzenburg** [13], **Hanna Lotter** [1*], **Iris Bruchhaus** [2,14*]

**1** RG Molecular Infection Immunology, Bernhard Nocht Institute for Tropical Medicine, Hamburg, Germany, **2** RG Host-Parasite Interaction, Bernhard Nocht Institute for Tropical Medicine, Hamburg, Germany, **3** Institute for Infection Research and Vaccine Development, Center for Internal Medicine, University Medical Center Hamburg-Eppendorf, Hamburg, Germany, **4** Department for Clinical Immunology of Infectious Diseases, Bernhard Nocht Institute for Tropical Medicine, Hamburg, Germany, **5** German Center for Infection Research (DZIF), Partner Site Hamburg-Lübeck-Borstel-Riems, Hamburg, Germany, **6** Cellular Parasitology Department, Bernhard Nocht Institute for Tropical Medicine, Hamburg, Germany, **7** Arbovirus and Entomology Department, Bernhard Nocht Institute for Tropical Medicine, Hamburg, Germany, **8** Laboratory of Mass Spectrometry, BIOCEV, Faculty of Science, Charles University in Prague, Prague, Czech Republic, **9** RG Protozoa Immunology, Bernhard Nocht Institute for Tropical Medicine, Hamburg, Germany, **10** I. Department of Medicine, University Medical Center Hamburg-Eppendorf, Hamburg, Germany, **11** Division of Innate Immunity, Research Center Borstel, Leibniz Lung Center (Airway Research Center North), German Centre for Lung Research, Borstel, Germany, **12** Department of Infection Epidemiology, Bernhard Nocht Institute for Tropical Medicine, Hamburg, Germany, **13** Institute of Clinical Molecular Biology, University of Kiel, Kiel, Germany, **14** Department of Biology, University of Hamburg, Hamburg, Germany

☯ These authors contributed equally to this work.
\* barbara.honecker@bnitm.de (BH); lotter@bnitm.de (HL); bruchhaus@bnitm.de (IB)

## Abstract

The parasitic protozoan *Entamoeba histolytica* secretes extracellular vesicles (EVs), but so far little is known about their function in the interaction with the host immune system. Infection with *E. histolytica* trophozoites can lead to formation of amebic liver abscesses (ALAs), in which pro-inflammatory immune responses of Ly6C[hi] monocytes contribute to liver damage. Men exhibit a more severe pathology as the result of higher monocyte recruitment and a stronger immune response. To investigate the role of EVs and pathogenicity in the host immune response, we studied the effect of EVs secreted by low pathogenic *Eh*A1 and highly pathogenic *Eh*B2 amebae on monocytes. Size and quantity of isolated EVs from both clones were similar. However, they differed in their proteome and miRNA cargo, providing insight into factors potentially involved in amebic pathogenicity. In addition, EVs were enriched in proteins with signaling peptides compared with the total protein content of trophozoites. Exposure to EVs from both clones induced monocyte activation and a pro-inflammatory immune response as evidenced by increased surface presentation of the activation marker CD38 and upregulated gene expression of key signaling pathways (including NF-κB, IL-17 and TNF signaling). The release of pro-inflammatory cytokines was increased in EV-stimulated monocytes and more so in male- than in female-derived cells. While *Eh*A1 EV stimulation caused elevated myeloperoxidase

**Data availability statement:** The mass spectrometry proteomics data have been deposited to the ProteomeXchange Consortium via the PRIDE partner repository (https://www.ebi.ac.uk/pride/) with the dataset identifier PXD059636. miRNA and bulk RNA sequencing data are accessible from NCBI Sequence Read Archive (SRA) under BioProject IDs PRJNA1201928 and PRJNA1202027, respectively.

**Funding:** This work was supported by funding from the Joachim Herz Foundation (IB, HL) (https://www.joachim-herz-stiftung.de/en/) and Deutsche Forschungsgemeinschaft (grant number BR 1744/17-2 (IB) and research unit 5068 - Sex differences in immunity (grant number LO 1426/4-1 (HL)). The funders did not play any role in study design, data collection and analysis, decision to publish or preparation of the manuscript. Next-Generation-Sequencing for miRNA analysis was carried out at the Competence Centre for Genomic Analysis (Kiel), which is supported by the DFG Research Infrastructure NGS_CC (project 407495230) as part of the Next Generation Sequencing Competence Network (project 423957469).

**Competing interests:** The authors have declared that no competing interests exist.

(MPO) release by both monocytes and neutrophils, *Eh*B2 EV stimulation did not, indicating the protective role of MPO during amebiasis. Collectively, our results suggest that parasite-released EVs contribute to the male-biased immunopathology mediated by pro-inflammatory monocytes during ALA formation.

## Author summary

Parasites communicate with their host via small membranous extracellular vesicles (EVs) that can shuttle cargo and thus information between cells. The protozoan parasite *Entamoeba histolytica* releases EVs but not much is known about their role in the interaction with the host immune system. Infection with *E. histolytica* can lead to amebic liver abscess (ALA) formation. Innate immune cells, particularly monocytes, contribute to liver damage by releasing microbicidal factors. Men have a more severe ALA pathology as the result of a stronger monocyte immune response. In this study, we analyzed the effect of EVs from differently virulent *E. histolytica* clones on monocytes to better understand their interaction. EVs of both clones were similar in size and quantity but differed in their cargo, which provides information on factors potentially involved in pathogenicity. Monocytes responded to EVs of both clones in a pro-inflammatory manner that reflected the immune processes occurring during ALA *in vivo*, including the bias towards the male sex. Only EVs of amebae with low pathogenicity, and not those released by the highly pathogenic clone, elicited secretion of the granular enzyme myeloperoxidase, which plays a protective role during ALA. Overall, our data suggest that EVs may contribute to liver injury.

## Introduction

It is well established that the communication between parasites and the host immune system involves extracellular vesicles (EVs) derived from both pathogen and host cells, which transfer cargo, such as proteins and RNA, and thus information from one to the other [1]. EVs are membranous vesicles released either by fusion of multivesicular bodies from the endolysosomal pathway with the plasma membrane or by direct budding of the plasma membrane into the extracellular space [2]. Depending on the infection context, parasitic EVs may play a role in parasite persistence or clearance and EVs inducing protective immune responses are studied as putative vaccine candidates [3–5].

The protozoan parasite *Entamoeba histolytica* is the causative agent of amebiasis, a disease endemic to tropical areas of the world that is responsible for significant disease burden and an estimated 26,000 deaths annually according to data from the Global Burden of Disease Study 2016 [6]. Infection occurs predominantly through the ingestion of food or water contaminated with fecal matter and most infections remain asymptomatic [7,8]. As the result of yet unidentified triggers, *E. histolytica* can become invasive, causing amebic dysentery or colitis in the intestine and, in some instances, disseminating to the liver via the portal vein, where amebic liver abscesses (ALAs) are formed. ALAs occur in about 1% of amebiasis cases and are lethal if left untreated [9]. ALA formation is heavily biased towards adult men despite similar infection rates between the sexes [10–12] and mediated mainly by infiltrating pro-inflammatory monocytes, but also neutrophils recruited to the site of infection [13–16]. Depletion of monocytes and neutrophils in a mouse model for hepatic amebiasis was shown to reduce abscess size [13]. In this ALA model, classical Ly6C$^{hi}$ monocytes contribute to liver

injury by releasing pro-inflammatory cytokines such as TNF and reactive oxygen species (ROS) [13,15]. Abscesses are larger in male mice, mainly due to higher monocyte recruitment via CCL2-CCR2 signaling to the liver in males compared with females, as well as increased release of pro-inflammatory cytokines in males [17,18]. The recruitment of monocytes and neutrophils to the liver was shown to be dependent on testosterone [16,18].

Only few studies on *E. histolytica* EVs have been performed to date [19–21]. In macrophages derived from the THP-1 cell line, *Eh*EVs were shown to dampen polarization into type 2 macrophages and to modulate the metabolism as well as release of pro- and anti-inflammatory cytokines [21]. In human neutrophils, pretreatment with EVs of this parasite was found to inhibit amebae-induced oxidative burst and neutrophil extracellular trap (NET) formation, indicating an immunosuppressive effect of the EVs [20]. For the reptile pathogen *Entamoeba invadens*, there is evidence that EVs may play a role in the communication among amebae regarding transformation between trophozoite and cyst stages during their life cycle [19].

Here, we studied the interaction between *E. histolytica* EVs and murine monocytes, which are of particular interest due to their pivotal role in ALA immunopathology, and also assessed myeloperoxidase (MPO) release by EV-stimulated neutrophils. We used EVs from two clones differing in their capacity to induce abscess formation in the mouse model – the low pathogenic *Eh*A1 clone and the highly pathogenic *Eh*B2 clone [22]. Since these clones constitute a unique approach and provide insight into factors involved in amebic pathogenicity, we included a comprehensive characterization of *Eh*A1- and *Eh*B2-derived EVs and their cargo in this study.

Using proteomics, we determined the protein constitution of these EVs as well as whole trophozoites. Additionally, we analyzed the EV miRNA content and bioinformatically predicted novel *E. histolytica* miRNAs. When stimulating murine monocytes, *E. histolytica*-derived EVs of both clones induced pro-inflammatory immune responses. Our results suggest that monocyte-mediated immunopathology during hepatic amebiasis may be driven by parasitic EVs.

## Results

### Trophozoites of differently pathogenic *E. histolytica* clones release EVs similar in size and morphology

To assess particle release from the different parasite clones and size distribution of these particles, EVs were isolated from *E. histolytica*-conditioned medium using differential ultracentrifugation (Fig 1A) and subsequently subjected to nanoparticle tracking analysis (NTA) (Fig 1B–E). The majority of released particles ranged in size between 80 and 400 nm (Fig 1B and 1C). EVs released by clones *Eh*A1 and *Eh*B2 did not exhibit significant differences in terms of modal size (Fig 1D) or abundance (Fig 1E). The heterogeneity of vesicle sizes determined by NTA could also be visualized by transmission electron microscopy (TEM) (Fig 1F). Using immuno-gold labeling, we detected the presence of *E. histolytica* galactose/N-acetylgalactosamine (Gal/GalNAc) lectin and lipopeptidophosphoglycan (LPPG) on the surface of both *Eh*A1 and *Eh*B2 EVs, thus verifying the amebic origin of the visualized particles (Fig 1F).

### *Eh*A1 and *Eh*B2 EVs contain proteins associated with virulence and exhibit differences between the clones that are distinct from those in trophozoites

To characterize the protein composition of *Eh*A1 and *Eh*B2 EVs and compare them to the protein composition of whole trophozoites, we performed liquid chromatography-mass spectrometry. Together, a total of 889 proteins were detected in the *Eh*A1 and *Eh*B2 EV proteomes (Fig 2A). 852 different proteins were identified in a minimum of 2 out of 3 *Eh*A1 EV samples and 703 proteins in 2 out of 3 *Eh*B2 EV samples (Fig 2A and 2C). 666 proteins were common to both EV proteomes (Fig 2A), of which only 1, a hypothetical protein with unknown

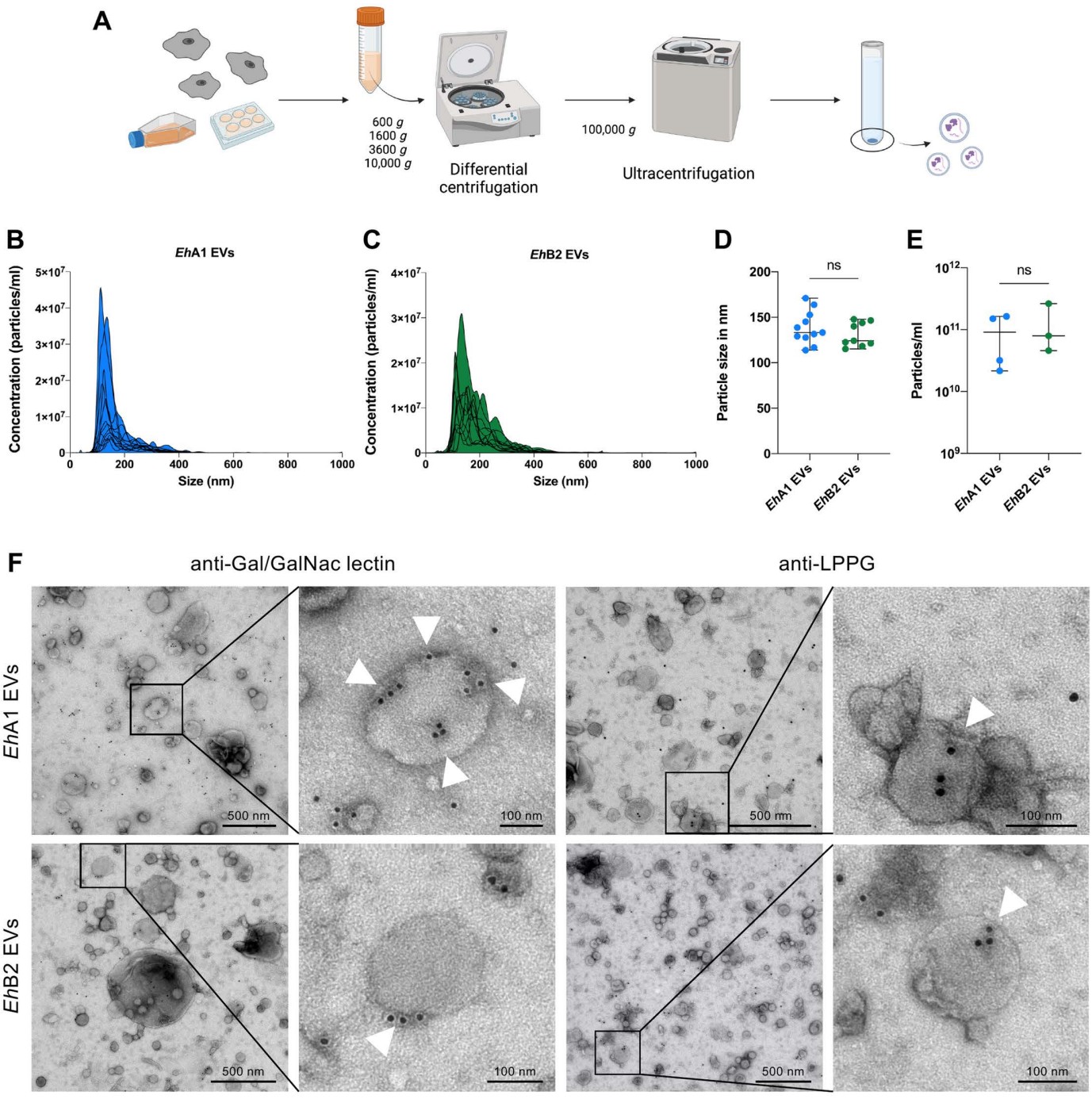

**Fig 1. Physical characterization of *E. histolytica* EVs. (A)** Workflow used for the isolation of EVs from *E. histolytica*-conditioned medium. Differential centrifugation of supernatants was performed at increasing speeds for 15 min each to clear the samples of cellular debris. EVs were pelleted by ultracentrifugation at 100,000 *g* for 1 h and washed once with filtered PBS prior to further use. Created in BioRender. **(B, C)** Nanoparticle tracking measurements of *Eh*A1 EVs (B) and *Eh*B2 EVs **(C)** (data averaged from 5 videos per sample, overlay of multiple independent measurements, n = 12–13). **(D)** Comparison of the modal particle size of *Eh*A1 EVs and *Eh*B2 EVs as determined by nanoparticle tracking (n = 9–11, unpaired *t* test, ns = not significant). **(E)** Comparison of the particle concentration of *Eh*A1 and *Eh*B2 EV samples in a set of standardized experiments (n = 3-4, unpaired *t* test, ns = not significant). **(F)** Transmission electron microscopic visualization of *Eh*EVs subjected to immunogold labeling with rabbit anti-galactose/N-acetylgalactosamine (Gal/GalNAc) lectin or mouse anti-lipopeptidophosphoglycan (LPPG) primary and gold-conjugated secondary antibodies. Arrowheads indicate sites of labeling on individual EVs. Shown are representative images of one sample for *Eh*A1 and *Eh*B2 each.

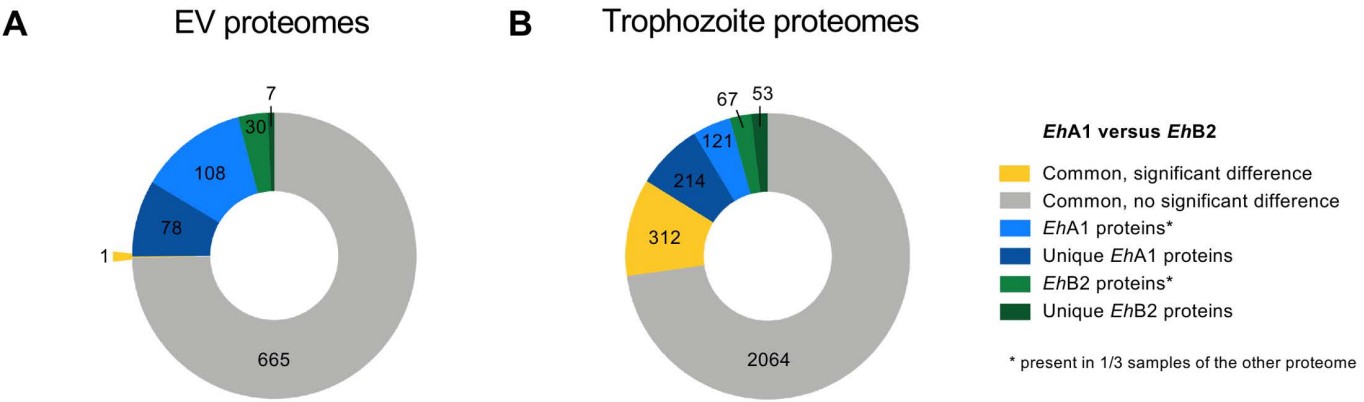

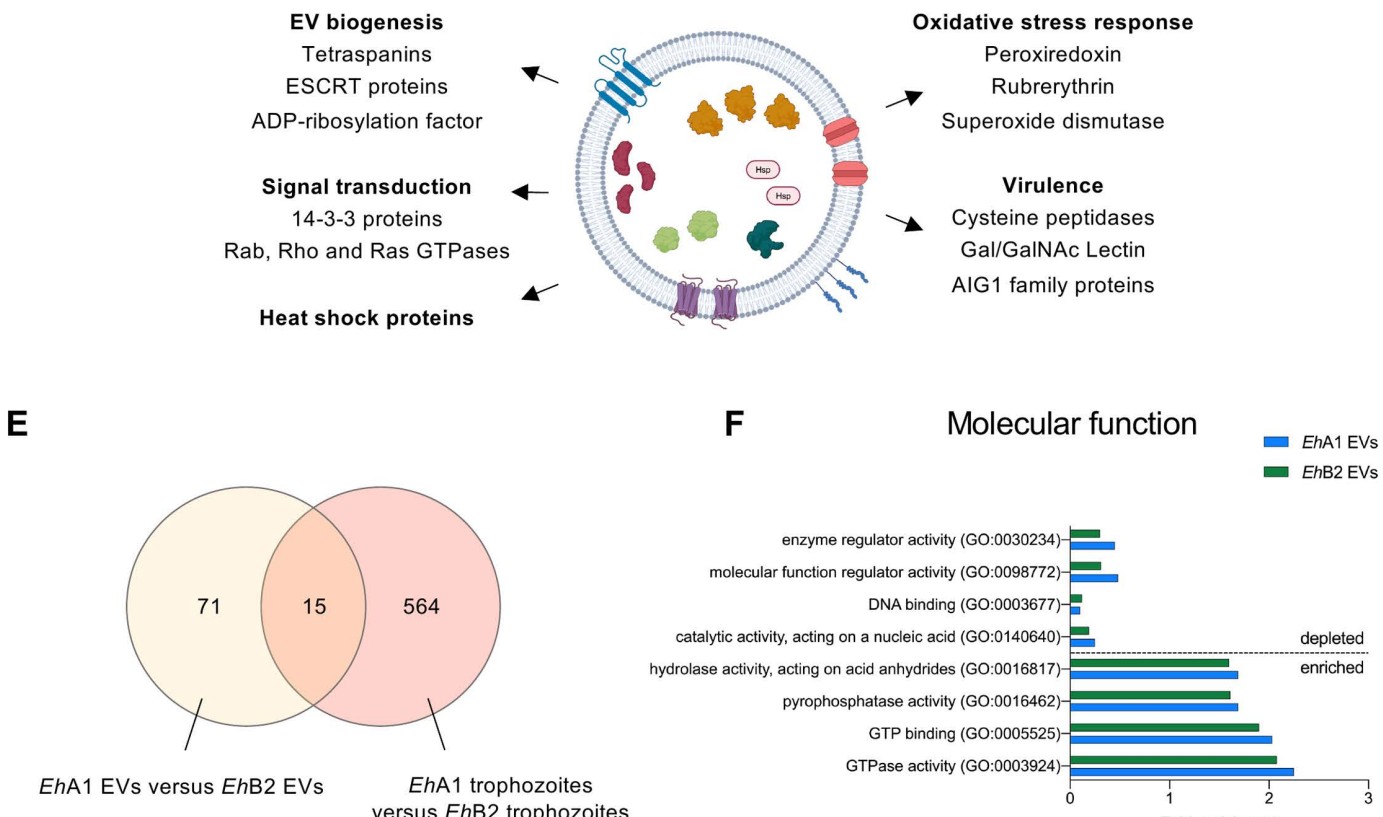

**Fig 2. Comparison of the *E. histolytica* vesicle and trophozoite proteomes. (A, B)** Quantitative comparison of *Eh*A1 and *Eh*B2 EV (A) and trophozoite (B) proteomes. Shown are proteins significantly differentially abundant between two proteomes (FDR $p < 0.05$, s0 = 0.5, fold change $\geq |2|$) (yellow) as well as proteins present in only one of both proteomes (blue/ green). Proteins were considered to be a part of the respective proteome if they were present in at least 2/3

samples. Hence, we differentiated between proteins completely unique to one proteome (absent from all samples of the other proteome) and proteins belonging to only one proteome but not unique (present in 1/3 samples of the other proteome). **(C)** Overview of the total number of proteins, hypothetical proteins, proteins with a transmembrane (TM) domain or signal peptide in each of the analyzed proteomes based on annotations in AmoebaDB release 68 [23]. **(D)** Schematic overview of the *E. histolytica* EV proteome with selected proteins of interest. Created in BioRender. **(E)** Venn diagram depicting the number of proteins differentially abundant in *Eh*A1 compared with *Eh*B2 EVs, as well as proteins differentially abundant in *Eh*A1 compared with *Eh*B2 trophozoites and amount of proteins common to both datasets. **(F)** Selected molecular function GO terms associated with proteins enriched or depleted in EV proteomes compared with trophozoite proteomes, based on statistical overrepresentation test performed with Panther knowledgebase [27,28].

function, was significantly differently abundant (fold change ≥ |2|, FDR $p < 0.05$, s0 = 0.5) (Fig 2A and S1 Table). 78 proteins were completely unique to *Eh*A1, while 7 proteins were exclusive to *Eh*B2 EVs (Fig 2A and S1 Table), with 29 and 4 of these, respectively, listed in AmoebaDB [23] as hypothetical or predicted proteins (S1 Table). Molecular function gene ontology (GO) term analysis of the unique *Eh*A1 EV proteins revealed an association of these proteins with transporter and GTPase signaling activity (S2 Table), consistent with the exclusive presence of multiple GTPases in this EV proteome (S1 Table).

With regard to trophozoite proteomes, 2711 proteins were detected in *Eh*A1 and 2496 in *Eh*B2 (Fig 2B and 2C and S4 Table). A total of 2376 of these were common to trophozoites of both clones, of which 312 significantly differed in their abundance between *Eh*A1 and *Eh*B2 (fold change ≥ |2|, Fig 2B and S4 Table). 214 proteins were detected uniquely in *Eh*A1 and 53 proteins uniquely in *Eh*B2 trophozoite proteomes (Fig 2B and S4 Table).

The relative amount of proteins with a transmembrane domain was higher in EV proteomes (31.10% in *Eh*A1, 33.85% in *Eh*B2) than in trophozoite proteomes (13.32% in *Eh*A1, 13.3% in *Eh*B2) (Fig 2C). Similarly, EV proteomes were enriched in proteins with a signal peptide (27.11% in *Eh*A1, 29.16% in *Eh*B2) compared to their trophozoite counterparts (9.15% in *Eh*A1, 10.10% in *Eh*B2) (Fig 2C). Proteins present in EV proteomes included those with a function in EV biogenesis, such as endosomal sorting complex required for transport (ESCRT) proteins [24] and tetraspanins (TSPAN1, TSPAN4, TSPAN12) [25] (Fig 2D and S5 Table). Additionally, proteins involved in signal transduction, heat shock, oxidative stress response and virulence were detected (Fig 2D and S1 Table). To determine how similar *E. histolytica* EV proteomes were to those of other organisms, we analyzed the presence of amebic orthologs for the top 100 EV proteins according to Vesiclepedia [26] in our data. We identified amebic orthologs for 63 of the top 100 EV proteins, 44 of which were present in *Eh*A1 EVs and 42 in *Eh*B2 EVs (S6 Table).

To assess whether differences between *Eh*A1 and *Eh*B2 trophozoites were conserved in their corresponding EVs, we compared significantly different proteins between EV and trophozoite proteomes. For this, we considered all proteins significant that were differentially abundant with fold change ≥ |2| and FDR $p < 0.05$ or completely unique to one clone, amounting to 86 proteins of interest for EV proteomes and 579 for trophozoite proteomes. We found that only 15 of these differentially abundant proteins were shared (Fig 2E). Notably, of the 71 proteins different between *Eh*A1 and *Eh*B2 EV proteomes but not trophozoite proteomes, 15 were unique to EVs (S1 Table). Regarding the remaining 56 proteins, this result may indicate a selectivity for protein packaging into EVs that differs between the two clones.

A statistical overrepresentation test (Panther knowledgebase [27,28]) revealed that proteins depleted in both *Eh*A1 and *Eh*B2 EVs compared to their cells of origin were associated with the nucleus and organelle lumen, and have functions in DNA binding and catalytic activity acting on a nucleic acid (Figs 2F and S2 and S7 Table). Proteins enriched in EVs of both clones were predominantly membrane-associated and involved in hydrolase or pyrophosphatase activity, GTP binding and GTPase activity (Figs 2F and S2 and S7 Table).

Taken together, *E. histolytica* EVs share key proteins with EVs of other organisms, including typical EV markers and key biogenesis proteins such as tetraspanins. While *Eh*A1 and *Eh*B2 EVs showed distinct protein compositions, these differences largely did not reflect the proteins differentially abundant between *Eh*A1 and *Eh*B2 trophozoites.

### *E. histolytica* EVs contain previously unidentified miRNAs

To analyze the micro RNAs (miRNAs) present in *Eh*A1 and *Eh*B2 EVs and evaluate their potential in the mediation of gene expression when transferred between cells, we isolated total EV RNA and performed miRNA sequencing. Out of all sequence reads, between 5,848,946 and 20,900,954 unique sequences per sample were detected. Curiously, none of the 199 previously described *E. histolytica* miRNAs were present in the sequencing data, with the exception of low amounts of Ehi-miR-4 [29]. Moreover, we only detected 26 out of 140,943 small RNA (sRNA) sequences published by Zhang *et al.* [30], most of them in very low abundance (S8 Table). Hence, to identify the miRNAs sequenced in our EV samples, we performed *de novo* miRNA prediction using BrumiR algorithm [31]. 1016 different miRNA sequences were identified overall, of which 167 were present in at least 2 out of 3 samples of either *Eh*A1 or *Eh*B2 with a minimum of 5 counts (S9 Table). We defined these 167 miRNA sequences as our EV miRNA dataset and, following the nomenclature established by Mar-Aguilar *et al.* [29], named them Ehi-miR-200 to Ehi-miR-366. All identified sequences were unique and did not exhibit identity with previously described *E. histolytica* sRNAs or miRNAs [29,30,32]. All 167 miRNAs were present in *Eh*A1 EVs, while only 165 were detected in *Eh*B2 EVs (Fig 3B and S9 Table). Out of the remaining 2, the difference in miRNA expression between *Eh*A1 and *Eh*B2 EVs was statistically significant (fold change ≥ |2|, FDR $p < 0.05$) only for Ehi-miR-200 (Fig 3F and S9 Table).

To elucidate the potential of these miRNAs to regulate gene expression, we performed target gene prediction using miRanda algorithm [33,34]. Since EVs are not only relevant in the communication between a parasite and its host, but also in inter-parasite communication, we performed target gene prediction for both the human genome (as the natural host) and the *E. histolytica* genome. Individual miRNAs were found to have anywhere between 350 and 10,350 potential targets in the human genome (Fig 3B) and between 5 and 1355 potential targets in the *E. histolytica* genome (Fig 3C). For most putative targets in the human genome, less than 20 different miRNAs exhibited the capacity to bind, while in some instances, more than 50 different miRNAs were detected to possibly bind to the same target (Fig 3D). In the *E. histolytica* genome, most identified targets were associated with less than 10 different miRNAs (Fig 3E). Considering that all putative targets of all 167 miRNAs covered practically the entire human and *E. histolytica* genomes, we subsequently focused the target analysis on individual miRNAs of interest (S10 Table). For Ehi-miR-200, which was significantly differentially expressed between *Eh*A1 and *Eh*B2 EVs, a total of 2480 targets in 1227 different genes were identified in the human genome (S10 Table). This corresponds to about 6% of all human protein-coding genes [35,36]. Molecular function GO terms associated with these targets include 'calmodulin binding', 'guanyl-nucleotide exchange factor activity' and others (Fig 3G) that indicate a potential of Ehi-miR-200 to mediate intracellular signaling cascades in host target cells.

In summary, we here identified novel *E. histolytica* miRNAs present in EVs that have the potential to regulate gene expression in host and amebic target cells. Two of these miRNAs differed in their expression between *Eh*A1 and *Eh*B2 EVs.

### Primary monocytes internalize and are activated by *E. histolytica* EVs

To track cellular uptake of EVs by murine bone marrow-derived monocytes, we labeled *Eh*A1 and *Eh*B2 EVs with the dye BCECF,AM and evaluated fluorescence in stimulated cells via flow cytometry (Figs 4A and S3B). The majority of male- and female-derived monocytes were

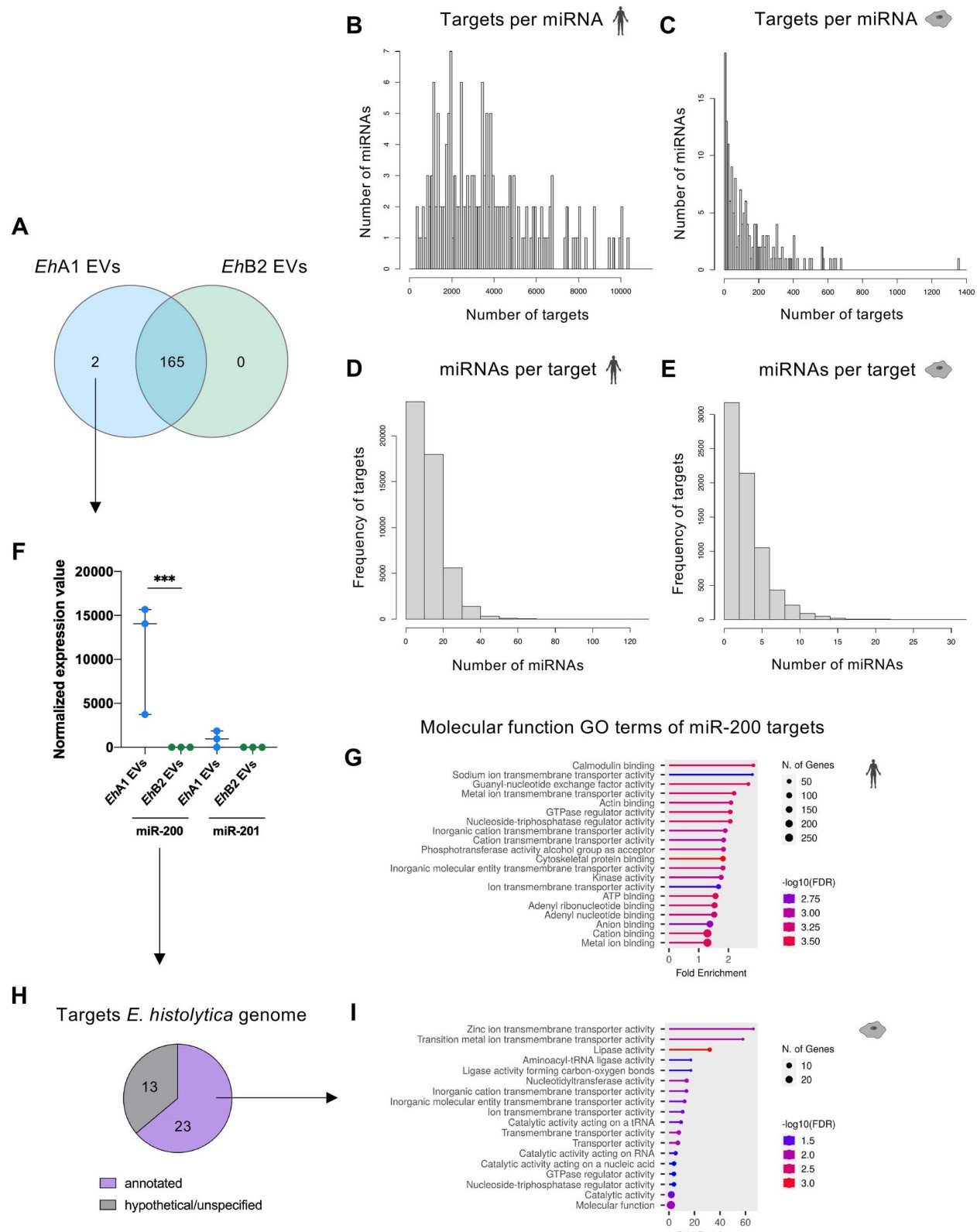

**Fig 3. Analysis of novel miRNAs in *E. histolytica* EVs. (A)** Comparison of the number of miRNAs detected in *Eh*A1 compared with *Eh*B2 EVs, based on *de novo* miRNA prediction using BrumiR algorithm version 3.0 [31] (n = 3 for each clone). **(B–E)** Quantitative analysis of potential

targets of novel *E. histolytica* miRNAs in the *Homo sapiens* and *E. histolytica* genome, predicted using miRanda algorithm version 3.3a [33,34]. Shown are the amount of potential targets per miRNA in the *H. sapiens* (B) and *E. histolytica* (C) genome, as well as the amount of miRNAs potentially binding to the same target (D and E). **(F)** Normalized expression values of the two miRNAs detected only in *Eh*A1 and not *Eh*B2 EVs (EDGE test, *** FDR *p* < 0.001, n = 3). **(G)** Molecular function GO terms associated with identified miR-200 targets in the human genome (analysis performed with shinyGO version 80 [37], shown are the top 20 GO terms). **(H)** Amount of annotated versus hypothetical or unspecified predicted targets of miR-200 in the *E. histolytica* genome (according to AmoebaDB version 68 [23]). **(I)** Molecular function GO terms associated with miR-200 targets in the *E. histolytica* genome with an annotated function (analysis performed with shinyGO version 80).

positive for BCECF following stimulation with labeled *Eh*A1 or *Eh*B2 EVs (Fig 4B and 4C). Both the percentage of positive cells and the median fluorescence intensity (MFI) were higher in EV-stimulated cells compared with mock controls despite some fluorescence detected also in these controls (Fig 4B–D). These results indicate a specific absorption of EVs by monocytes.

We next examined the effect of EV stimulation on the presence of the monocyte activation marker CD38 and C-C chemokine receptor type 2 (CCR2) on the cell surface using flow cytometry. CD38 is an enzyme catalyzing the synthesis of the second messenger adenosine diphosphate ribose (ADPR) and its cyclic form cADPR and thus mediates multiple functions during cell activation through calcium signaling [38,39]. CCR2 is a receptor for the chemokine CCL2, which is released during inflammation [40]. As such, CCR2 is crucial for monocyte egress from the bone marrow and recruitment to sites of infection [41].

Here, we combined data from male- and female-derived monocytes as no significant differences between the sexes could be detected. We applied the differentiation into classical pro-inflammatory monocytes expressing high amounts of Lymphocyte antigen 6 C (Ly6C$^{hi}$ monocytes) and non-classical Ly6C$^{lo}$ monocytes, which are considered reparative monocytes [42,43] (Figs 4F, 4G and S4). The majority of isolated monocytes across all experiments were Ly6C$^{hi}$ monocytes (53.9% - 90.7% of CD11b$^+$ cells) and only a minority Ly6C$^{lo}$ monocytes (5.1% - 33.2% of CD11b$^+$ cells) (S4C and S4D Fig). *Eh*A1 EVs and *Eh*B2 EVs both significantly increased the percentage of CD38-positive Ly6C$^{hi}$ monocytes (median 18.7% (*Eh*A1), 20.65% (*Eh*B2)) compared with mock controls (median 3.8%) (Fig 4E and 4F) without affecting the viability of stimulated cells (S6B Fig). The same was observed for Ly6C$^{lo}$ monocytes, albeit to a lesser extent (median 7.1% (*Eh*A1) and 7.4% (*Eh*B2) CD38$^+$ cells versus 1.6% (mock control)) (Fig 4G). Heat inactivation of EVs, performed as a control for protein denaturation, led to a slight reduction in stimulatory capacity for *Eh*B2 EVs, but not *Eh*A1 EVs (Fig 4F and 4G). Our results on CD38 suggest that EVs released by both *E. histolytica* clones induce activation of monocytes. Although EV stimulation did not induce a change in the relative amount of CCR2$^+$ cells, a slight increase in the MFI for CCR2 could be observed on Ly6C$^{hi}$, but not Ly6C$^{lo}$ monocytes (S4E–H Fig). This finding hints at a stronger potential for migration of pro-inflammatory monocytes upon contact with *E. histolytica* EVs.

## EVs of differently pathogenic *E. histolytica* clones induce a pro-inflammatory gene expression profile in stimulated monocytes

After having shown that *E. histolytica* EVs can activate monocytes from male and female mice, we investigated the impact of EV stimulation on their gene expression profile by RNA sequencing and again combined data from male- and female-derived cells for this analysis. A total of 39 genes were significantly differentially expressed (fold change ≥ |2|, Bonferroni *p* < 0.05) between *Eh*A1 EV- and mock control-stimulated monocytes, all of which were upregulated in EV-stimulated cells (S11 Table and Fig 5A and 5B). In response to *Eh*B2 EV stimulation, 53 genes were upregulated in monocytes in comparison with mock controls, while 4 genes were downregulated (S11 Table and Fig 5A and 5C). No statistically significant differences in gene expression were detected between *Eh*A1 EV- and *Eh*B2 EV-stimulated cells (S11 Table). Genes

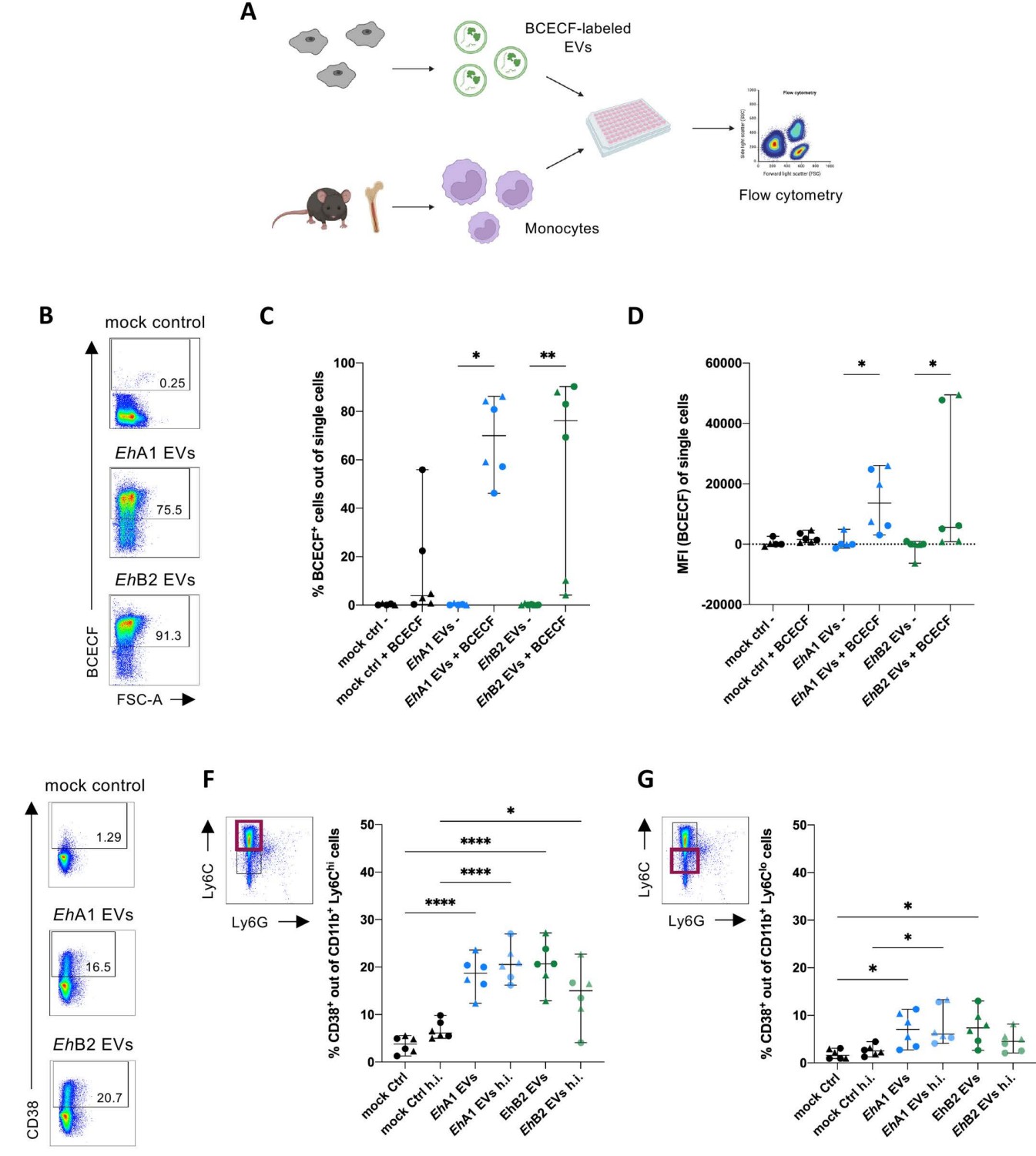

**Fig 4. Cellular uptake of *E. histolytica* EVs and activation of primary monocytes.** **(A)** Graphical depiction of the workflow used for the determination of EV uptake by primary monocytes. Bone-marrow derived monocytes from male (shown as dots in graphs) and female (shown as triangles) mice were stimulated for 30 min with 0.5 μg EVs labeled with BCECF or corresponding volumes of controls and cellular uptake was quantified by spectral flow cytometry. Created in BioRender. **(B)** Dot plots of representative samples depicting flow cytometric identification of BCECF-positive cells following stimulation with mock control, *Eh*A1 EVs or *Eh*B2 EVs labeled with BCECF. **(C)** Percent BCECF+ monocytes after stimulation based on gates in **(B)**. **(D)** Median fluorescence intensity (MFI) of

BCECF in single cells after stimulation. MFIs were normalized to monocytes stimulated with unlabeled EVs for each independent experiment. (Kruskal-Wallis test with Dunn's multiple comparisons test, *$p < 0.05$, **$p < 0.01$, n = 6 from 3 independent experiments). (**E**) Dot plots of representative samples depicting surface expression of the activation marker CD38 on Ly6C<sup>hi</sup> monocytes upon stimulation. Monocytes were stimulated *in vitro* for 24 h with 1000 EVs/cell of *Eh*A1 or EhB2 EVs or corresponding volume mock control and subsequently analyzed by flow cytometry. (**F**) Percent CD38⁺ Ly6C<sup>hi</sup> monocytes following EV stimulation. To control for the effect of protein denaturation on stimulatory capacity, EV and control samples were heat inactivated (h.i.) at 95°C for 10 min prior to stimulation. (**G**) Percent CD38⁺ Ly6C<sup>lo</sup> monocytes following EV stimulation. (One-way ANOVA with Šídák's multiple comparisons test, *$p < 0.05$, ****$p < 0.0001$, n = 6).

induced by EV stimulation included those encoding for cytokines *Tnf, Cxcl2, Ccl3, Ccl4, Ccl5* and *Cxcl10* (S11 Table and Fig 5A–C). In accordance with this, most of the top Molecular function GO terms associated with genes upregulated upon EV stimulation were related to chemokine receptor binding, chemokine and chemoattractant activity (Fig 5D and 5E). Kyoto Encyclopedia of Genes and Genomes (KEGG) pathway analysis revealed that upregulated genes were involved in key immune signaling pathways, such as NF-κB signaling, TNF signaling and IL-17 signaling, among others (S5A and S5B Fig). Along with the induction of cytokine-encoding genes, several type I interferon-stimulated genes (ISGs) were upregulated upon EV stimulation, including *Icam1, Gbp5, Ifit1, Rsad2* and *Oasl1* (S11 Table and Figs 5A–C and S5). Furthermore, genes induced by EV stimulation included *Clec4e* encoding for the Mincle receptor, an important initiator of pro-inflammatory immune responses associated with phagocytosis [44], *Sod2*, involved in the processing of reactive oxygen species (ROS) and thus protection of cells from oxidative stress [45], and *Cd40*, which encodes a co-stimulatory molecule involved in the production of pro-inflammatory cytokines [46,47] (S11 Table and Fig 5A–C). The most significantly upregulated gene upon both *Eh*A1 and *Eh*B2 EV stimulation was *Acod1* (S11 Table and Fig 5B and 5C), encoding for Aconitate decarboxylase 1, an enzyme involved in the mediation of immune responses to inflammatory stimuli through the production of the metabolite itaconate [48]. The interferon regulatory factor *Irf4* was among the few genes downregulated upon EV stimulation (S11 Table and Figs 5A, 5C and S5D).

We validated the sequencing results using quantitative PCR (qPCR) for selected genes, showing increased expression of *Tnf, Cxcl2, Ccl5* and *Oasl1* upon both *Eh*A1 and *Eh*B2 EV stimulation compared with mock controls (Fig 5F–I). Interestingly, expression of *Cxcl2* was strongly increased in all monocytes cultured during stimulation, including mock controls, compared with pre-stimulation samples (Fig 5G, note the logarithmic scale).

These results show that stimulation of monocytes with *E. histolytica*-derived EVs led to the induction of a predominantly pro-inflammatory immune response on mRNA level, primarily characterized by upregulation of chemokine-encoding genes. Despite differences in pathogenicity of the amebic clones, EVs released by *Eh*A1 and *Eh*B2 did not trigger significantly different immune responses in this context.

## Release of pro-inflammatory cytokines is enhanced by EV stimulation and stronger in male- than female-derived monocytes

Several studies have previously underlined the important role of cytokine signaling in ALA immunopathology [13,14,18]. Moreover, monocytes from male mice were shown to release higher amounts of TNF and CXCL1 during hepatic amebiasis compared with their female counterparts, contributing to the differing disease burdens observed between the sexes [18]. To assess cytokine release in response to *E. histolytica* EVs, enzyme-linked immunosorbent assay (ELISA) and a bead-based multiplex assay (LEGENDplex) were performed on supernatants of stimulated monocytes. Here, we investigated monocytes from male and female mice separately. We found that *Eh*A1 and *Eh*B2 EV stimulation resulted in marked increases in the release of IL-12p40, CXCL1, IL-6, TNF, IL-1β and CCL3 by monocytes in comparison

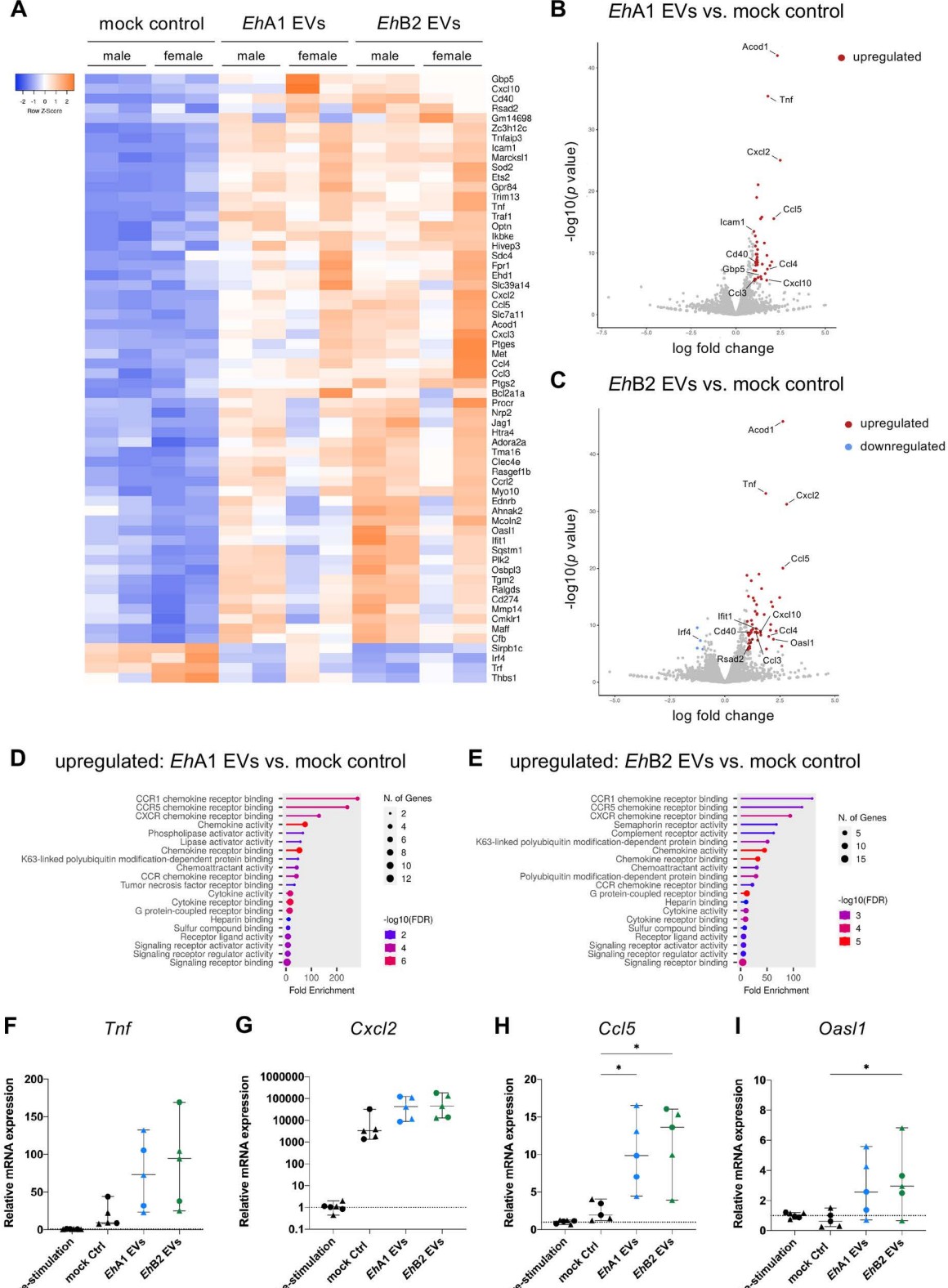

**Fig 5. Transcriptional profile of EV-stimulated monocytes.** Male and female bone marrow-derived monocytes were stimulated for 8 h *in vitro* with 1000 EVs/cell or equal volume mock control. mRNA expression levels were subsequently analyzed using RNA sequencing. (**A**) Heatmap depicting expression levels of genes significantly differentially expressed between *Eh*A1 EV and/or *Eh*B2 EV-stimulated

monocytes and mock controls (fold change ≥ |2|, Bonferroni-corrected $p < 0.05$, n = 4). **(B, C)** Volcano plots depicting relative gene expression between *Eh*A1 EVs (B) or *Eh*B2 EVs (C) and mock controls. Genes significantly upregulated in EV-stimulated monocytes are depicted in red, whereas genes significantly downregulated are depicted in blue (log fold change ≥ |1|, Bonferroni-corrected $p < 0.05$, n = 4). Selected genes of interest are labeled. **(D, E)** Molecular function GO term analysis of significantly upregulated genes in *Eh*A1 EV (D) or *Eh*B2 EV (E)-stimulated monocytes compared with mock controls (analysis performed with shinyGO version 80 [37], shown are the top 20 GO terms). **(F, G, H, I)** qPCR analysis of selected genes of interest (based on A–C) in EV-stimulated and control male- (dots) and female-derived (triangles) monocytes. *Rps9* was used as calibrator. ΔCq values were normalized to the median Cq value of pre-stimulation controls. (One-way ANOVA with Dunnett's or Šídák's multiple comparisons test, *$p < 0.05$, n = 5-6).

with mock controls (Figs 6A and S6A–K). This was observed in monocytes derived from both sexes. However, for IL-12p40, CXCL1 and IL-1β, the effect was stronger in monocytes isolated from males than from females (Figs 6A and S6). Interestingly, heat inactivation of *Eh*B2 EVs led to a partial ablation of the observed effect, which was not observed upon heat inactivation of *Eh*A1 EVs (Figs 6A and S6A–K). Despite the fact that CCL2 is a main mediator of ALA immunopathology in mice [13] and is known to be expressed in the vast majority of Ly6C$^{hi}$ monocytes during ALA [18], we could not detect any CCL2 release by EV-stimulated monocytes.

### *Eh*A1 EVs but not *Eh*B2 EVs trigger increased myeloperoxidase release by monocytes and neutrophils

In addition to investigating cytokine release, we also analyzed release of the granular enzyme MPO in response to *E. histolytica* EVs. MPO is released from azurophilic granules during degranulation and extracellular trap release and catalyzes the formation of microbicidal mediators, thus contributing to the innate immune response to pathogen invasion [49,50]. MPO released by host cells has been previously shown to kill *E. histolytica* trophozoites by using parasite-derived hydrogen peroxide for the formation of cytotoxic hypochlorous acid [51].

Interestingly, MPO concentrations were higher in supernatants of *Eh*A1 EV-stimulated monocytes of both sexes than in mock controls, but not in supernatants of *Eh*B2 EV-stimulated cells (Fig 6B). Since neutrophils are known to be a major source of MPO in circulation [50] and also contribute to ALA, we decided to follow up on this intriguing discovery by stimulating neutrophils with *E. histolytica* EVs. When using bone marrow-derived neutrophils, no significant differences in MPO release between EV- and mock-stimulated cells could be detected (Fig 6C). However, when we isolated neutrophils from spleen and blood (peripheral neutrophils) and subjected them to EV stimulation, *Eh*A1 EVs triggered significantly increased MPO levels, while *Eh*B2 EVs did not (Fig 6D).

### Discussion

Parasite-derived EVs have been reported to modulate host immune responses to their benefit or disadvantage in the context of several infectious diseases before [1,3]. Here, we assessed the interaction of EVs released by the protozoan *E. histolytica* primarily with monocytes, but also neutrophils, which constitute key cell types in the innate immune response to infection with this parasite and contribute to pathology during ALA. Only a minority of *E. histolytica* infections lead to invasive disease and the underlying mechanisms are still largely unknown [8]. Use of the differently pathogenic *Eh*A1 and *Eh*B2 clones aids in the detection of factors and mechanisms involved in amebic pathogenicity, hence we analyzed EVs of both these clones.

We reported that *Eh*A1 and *Eh*B2 EV proteomes were enriched in transmembrane and signaling proteins compared with their parent cell proteomes, which is consistent with other EV proteomes [52,53], but not with a previously published *E. histolytica* EV proteome [19]. EV proteomes exhibited presence of many typical EV markers, including proteins involved

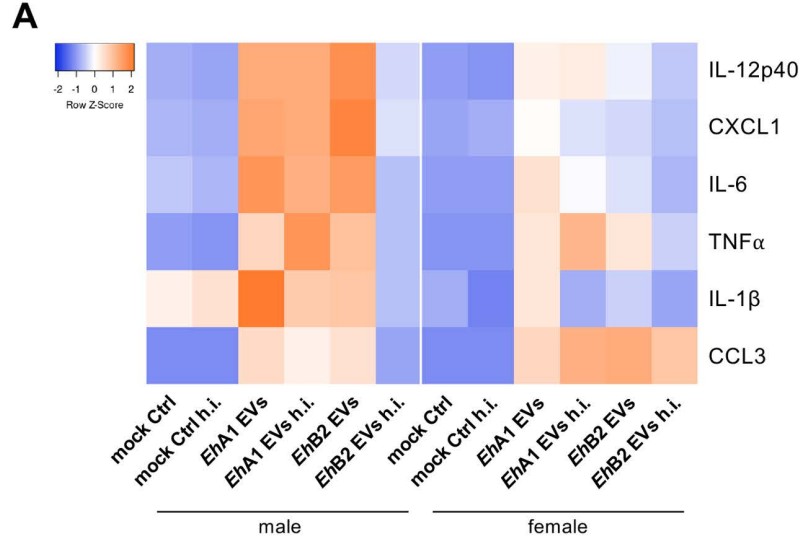

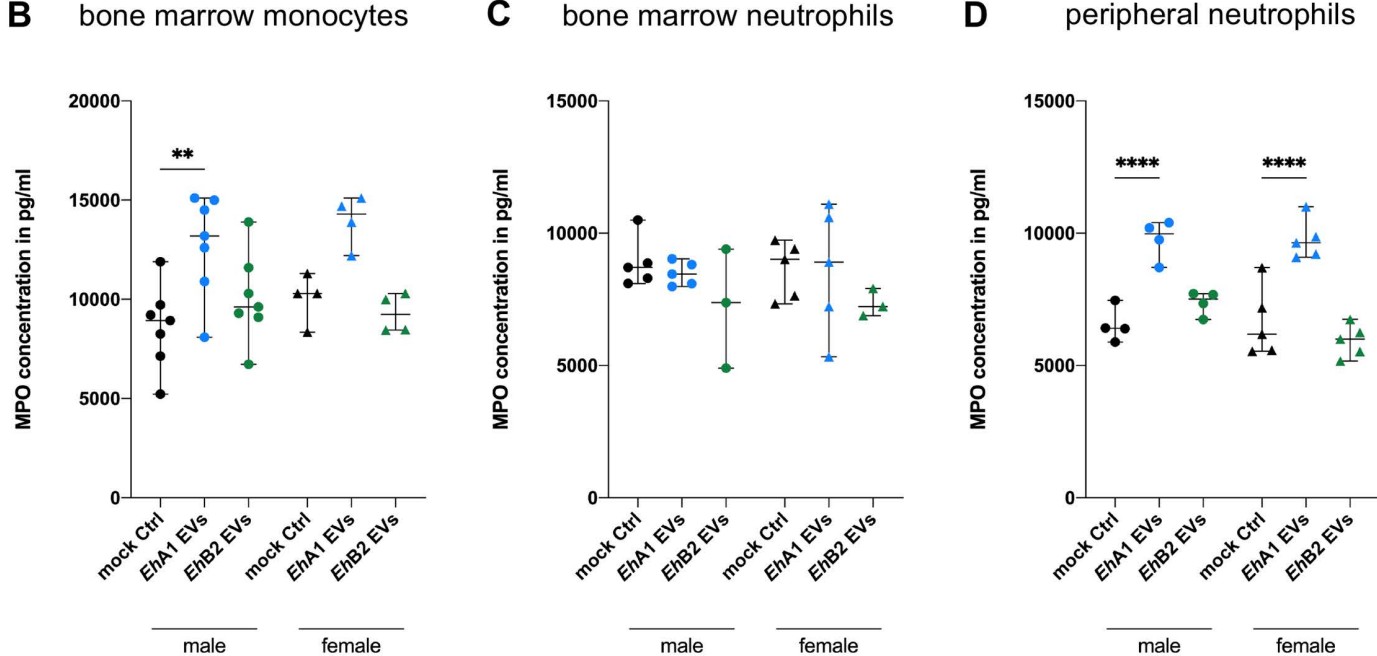

**Fig 6. Release of cytokines and myeloperoxidase upon EV stimulation.** Bone marrow-derived monocytes from male and female mice were stimulated for 24 h *in vitro* with 1000 EVs/cell or equal volume mock control. Supernatants of stimulated cells were analyzed by ELISA or flow cytometry-based multiplex cytokine assay (LEGENDplex). To control for the effect of protein denaturation on stimulatory capacity, EV and control samples were heat inactivated (h.i.) at 95°C for 10 min prior to stimulation. **(A)** Heatmap depicting median fluorescence intensity values for cytokines in supernatants of monocytes stimulated with EVs or controls as determined by LEGENDplex (IL-12p40, CXCL1, IL-6, TNF, IL-1β) or cytokine concentration as determined by ELISA (CCL3) (n = 3-6, depicted is the median of all samples of one condition). **(B, C)** Myeloperoxidase (MPO) concentration in supernatants of stimulated monocytes (B), bone marrow-derived neutrophils (C) and peripheral neutrophils (D) (isolated from spleen and blood, stimulated in the same manner as monocytes) as determined by ELISA. (One-way ANOVA with Šídák's multiple comparisons test, ** $p < 0.01$, **** $p < 0.0001$, n = 3-7).

in EV biogenesis. The detection of tetraspanins was particularly intriguing considering that previous proteome analyses of *E. histolytica* EVs had reported absence of this protein class and proposed that EV biogenesis may occur tetraspanin-independently in this parasite [19,20]. This hypothesis is refuted by our findings and suggests that EV biogenesis is more conserved in *E. histolytica* than previously thought. Another previous study reported increased CP activity in EVs released from *tspan4*-silenced *E. histolytica* compared with controls, suggesting that tetraspanins may be involved in controlling CP sorting into EVs and, consequently, a key amebic virulence mechanism [54]. Unique detection of tetraspanins in our study may result from the comprehensiveness of our compared to previous EV proteomes (889 proteins in contrast to 359 [19] and 597 [20] EV proteins). Furthermore, variances in the *E. histolytica* EV proteome constitution between these three studies can be explained by different experimental conditions impacting cargo packaging, such as different incubation times, culture media, or the use of the heterogeneous HM-1:IMSS amebic cell line versus *Eh*A1 and *Eh*B2 clones. Importantly, the method used for EV isolation critically influences their composition [55]. In contrast to our ultracentrifugation-based approach, EVs were precipitated chemically in the previous studies [19,20], resulting in different EV constitution.

When comparing EV proteomes of *Eh*A1 and *Eh*B2, we detected 86 significantly differentially abundant proteins, which may be interesting candidates to further investigate in the context of amebic pathogenicity as we currently still lack knowledge of many of their functions. The majority of these differentially abundant EV proteins were not also differentially abundant in corresponding trophozoite proteomes. We hypothesize that this is the result of a different selectivity for protein packaging between the two clones. Particularly these proteins as well as EV proteins that were not detected in trophozoite proteomes at all, and thus, must be highly enriched in EVs, should be considered for further analysis to better understand the role of EVs in host-parasite interaction and gain insight into mechanisms of pathogenicity.

In this work, apart from EV proteomes, we present the first whole trophozoite proteomes of *Eh*A1 and *Eh*B2 clones, which are among the largest and therefore most comprehensive *E. histolytica* proteomes available to date [56–60]. In light of the different pathogenic potential of these two clones, these proteomes will provide a valuable source of information to those studying mechanisms of amebic pathogenicity.

Next to protein cargo, EVs are known to also contain different RNA species. Of particular interest are miRNAs due to their ability to regulate gene expression in EV target cells [3]. Sharma *et al.* previously showed presence of RNA-induced silencing complex (RISC) proteins and selective packaging of sRNAs into *E. histolytica* EVs [19]. *E. histolytica* RISC proteins [61] were also found in the proteomes of our EVs (S1 Table). Furthermore, we identified novel *E. histolytica* miRNAs, which may modulate gene expression and intracellular signaling in host cells via EVs. Similar to what has been described for nematode EVs in mice [62], these miRNAs could suppress immune responses, but their gene-silencing effects require further validation.

During co-culture with primary bone marrow-derived monocytes, *Eh*A1 and *Eh*B2 EVs were taken up and induced an activated, pro-inflammatory phenotype, characterized by upregulated cytokine-encoding genes, ISGs, and other genes involved in key immune signaling pathways. This increased gene transcription also translated into elevated cytokine release on protein level. Several of these cytokines, namely IL-1β, IL-6, CCL3 and TNF, were also increasingly released by THP-1-derived macrophages after *Eh*EV stimulation in a previous study, in which EVs were also shown to be internalized by host macrophages [21].

Most interestingly, these observations largely mirrored previously described monocytic immune responses from *in vivo* ALA models. The activation marker CD38, which regulates cytokine release, phagocytosis and immune cell migration during inflammation [39],

was significantly increased in abundance on EV-stimulated monocytes and also on Ly6C<sup>hi</sup> monocytes during ALA in mice [15]. CD38-expressing monocytes contribute to immunopathology and constitute the major source of ROS during ALA, as demonstrated by the reduction in abscess size and monocyte infiltration in the liver in *Cd38*<sup>-/-</sup> mice [15]. Elevated CD38 amounts on Ly6C<sup>hi</sup> monocytes upon EV stimulation are concomitant with a generally pro-inflammatory immune response. Nonetheless, we herein also observed more CD38<sup>+</sup> Ly6C<sup>lo</sup> monocytes upon EV stimulation compared with controls. Ly6C<sup>lo</sup> monocytes, in contrast to their Ly6C<sup>hi</sup> counterparts, contribute to tissue repair during ALA in mice [14] and are recruited to the infected liver later than the rapidly recruited pro-inflammatory Ly6C<sup>hi</sup> monocytes [15]. Considering that CD38 mediates transendothelial migration of immune cells [39], it can be assumed that CD38 is induced in both monocyte subsets upon EV stimulation also for this purpose.

Additionally, several of the cytokines induced on mRNA and protein level upon EV stimulation have been shown to be elevated during ALA in mice, contribute to tissue damage, and lead to the recruitment of more immune cells to the site of infection. This includes CCL3 [13–15], TNF [13,18], CXCL1 [18] and IL-1β ([15], shown on mRNA level). In this context, EV stimulation experiments could also replicate the male-biased sex difference observed in cytokine secretion, which was previously reported especially for CXCL1, TNF and CCL2 [18]. Despite the pivotal role of CCL2 in monocyte recruitment and ALA immunopathology, we could not detect secretion of this cytokine in response to *E. histolytica* EVs here. Another cytokine induced by EV stimulation, IL-6, is also significantly elevated in serum of humans with ALA compared with asymptomatic carriers of *E. histolytica* [63].

Expression of ISGs is regulated by interferon (IFN) signaling and has been mainly studied in antiviral immunity, although some studies on ISGs in parasitic infection exist [64]. IFN signaling and ISG expression are mostly described to enhance viral elimination and thus play a protective role [65]. In contrast, the role of IFN signaling in parasitic disease so far is not well investigated and understood [64]. Upregulated ISG expression was previously studied by our group in the context of sex differences in neutrophils during ALA in mice [16]. ISGs that were upregulated in blood neutrophils during ALA in that study were also found to be induced upon EV stimulation in monocytes (*Rsad2, Oasl1, Ifit1*). In the ALA mouse study, the ISG response was stronger in neutrophils isolated from female compared with male mice, which is consistent with higher amebicidal and thus protective IFNγ activity in female mice [17,66,67]. We proposed that the less activated neutrophils with lower ISG expression in males may be less efficient in parasite killing, thus indirectly enabling liver destruction [16]. In our work presented here, we could not assess a sex difference in ISG expression in bulk RNA sequencing data due to the limited sample number, but this may be promising to analyze more in depth in the future. Research on ISGs in antiparasitic immunity, especially anti-amebic immunity, is currently scarce and clearly needed to better understand the complex interplay of ISG expression, monocytes and neutrophils, and their contribution to liver injury in hepatic amebiasis.

*Irf4,* which was downregulated upon EV stimulation, encodes an interferon regulatory factor known to negatively regulate toll-like receptor (TLR) signaling [68]. IRF4 furthermore promotes monocyte polarization into monocyte-derived dendritic cells or anti-inflammatory M2 macrophages [69–73]. Another downregulated gene, *Thbs1*, strongly correlated with M2 macrophages in a tumor context [74]. These data hint at a possible inhibition of monocyte polarization by *E. histolytica* EVs, particularly the polarization into an anti-inflammatory M2 macrophage phenotype. Supporting this hypothesis, stimulation of THP-1-derived macrophages with *Eh*EVs was previously shown to result in dampened polarization into the M2 type [21].

Interestingly, there were no significant differences in CD38 upregulation, cytokine release or the transcriptional responses of monocytes to *Eh*A1 and *Eh*B2 EV stimulation. As shown by TEM and proteomics, EVs of both clones contained many *E. histolytica* pathogenicity factors, such as LPPG and Gal/GalNAc lectin, which are known to induce pro-inflammatory responses in monocytes or macrophages. For example, exposure of human and murine macrophages to Gal/GalNAc lectin, which mediates parasitic adhesion to the host epithelium during intestinal invasion [75,76], results in increased NF-κB signaling and elevated TNF levels [77,78], as well as caspase-1 and NLRP3 inflammasome activation followed by elevated IL-1β release [79–81]. Activation of the NLRP3 inflammasome via TLR4 in macrophages also occurs in response to peroxiredoxin [82], which was present in EV proteomes of both clones. The glycan LPPG, which is part of the amebic glycocalyx that constitutes a physical barrier protecting the parasite from complement [83,84], is recognized by monocytes and macrophages via TLR2 and TLR4. This has been shown to induce NF-κB and, consequently, release of IL-6, IL-12p40, TNF and other cytokines [85,86]. LPPG was detected on the surface of both *Eh*A1 and *Eh*B2 EVs using immunogold labeling for TEM and can be assumed to account for some of the EV-induced cytokine secretion. Virulence factors such as Gal/GalNAc lectin and tissue-destructing CPs were previously found to be increased in EVs from *E. histolytica* in co-culture with neutrophils compared with EVs from amebae monoculture [20], indicating an active modulation of EV protein cargo during contact with host immune cells. In light of this finding, assessing EV composition after co-culture with monocytes compared with before would be useful in providing a more direct link between EV components and the observed effect in monocytes.

One of the main differences between monocyte immune responses to *Eh*A1 compared with *Eh*B2 EV stimulation in our study was the induction of MPO release by *Eh*A1 and not *Eh*B2 EVs. During amebic colitis, increased MPO levels are associated with tissue damage [87], hence, it may appear counterintuitive that EVs of the less pathogenic clone elicited higher MPO release by monocytes and neutrophils. However, a protective role of MPO activity during ALA has previously been described in the resistant Balb/c mouse model, wherein MPO inhibition resulted in less damage to amebae and larger abscesses [88]. In that same study, the susceptible hamster model exhibited lower MPO activity compared with the mouse model, allowing for the formation of ALAs in these hamsters. Increase of MPO activity by ascorbic acid treatment was shown to decrease abscess size in this hamster model [89]. These findings are corroborated by the results on EVs of differently pathogenic clones presented in this work. We suggest that MPO release might either be induced by antigens present on less pathogenic amebae and their EVs, resulting in efficient parasite clearance, or, conversely, actively blocked by more pathogenic amebae and their EVs. This notion is further supported by the fact that *mpo* expression was not induced in Ly6C[hi] monocytes upon infection with the highly pathogenic *Eh*B2 in the ALA mouse model [15]. Further work is needed to better understand the interplay of monocyte-derived MPO and amebae in the context of pathogenicity, and to what extent host- or parasite-derived factors play a role.

The observation that neutrophils isolated from murine bone marrow did not respond to EV stimulation with increased levels of MPO, but peripheral neutrophils did, corresponds to our previous observation of a more quiescent phenotype of bone marrow-derived neutrophils compared with a more activated one in circulating neutrophils in ALA mice [16]. *E. histolytica* EVs from resting parasites as well as EVs isolated from *E. histolytica*/ neutrophil co-cultures were previously shown to exert immunosuppressive effects on neutrophils *in vitro*, including the suppression of oxidative burst and a delay in NET release in response to treatment with live trophozoites [20]. The HM-1:IMSS strain of *E. histolytica* used for those experiments is generally pathogenic and the absence of elevated MPO release upon *Eh*B2 EV stimulation in

our work corresponds to their finding that HM-1:IMSS EVs alone did not induce ROS or NET release, as MPO is a crucial component of NET formation. To complete the picture on the role of neutrophils in the response to more or less pathogenic amebae or their EVs, analysis of ROS production and NETosis upon *Eh*A1 EV stimulation will be interesting, an aspect we did not include in the experiments presented here due to our focus on monocytes.

In addition to differential MPO induction, *Eh*A1 and *Eh*B2 EVs also differed in their susceptibility to heat inactivation, impacting their subsequent stimulatory potential. While the EV structure is generally stable during heat exposure [90] and miRNAs can withstand even boiling temperature [91], proteins are less resistant and denature under heat stress. Persistence of the *Eh*A1 EV stimulatory capacity even after heat inactivation combined with the presence of two miRNAs only in these and not *Eh*B2 EVs suggests that these miRNAs may be involved in the monocyte response. This finding together with the abovementioned prediction of Ehi-miR-200 to modulate intracellular signaling in human target cells reinforces the potential in studying these miRNAs more closely. Furthermore, the differentially abundant proteins between the EVs of both clones, specifically the ones more abundant in *Eh*A1, warrant further investigation in this context, as they may also partake in causing immune responses even in denatured form.

In conclusion, the pro-inflammatory response elicited by EV stimulation of monocytes and the parallels with known immune responses during ALA *in vivo* indicate that parasitic EVs may contribute to ALA immunopathology also in absence of direct contact of host cells with live amebae. However, it should be noted that we here used a simplified model in contrast to the complex interplay of different cells and the EVs they release during infection. Just as EV composition in the liver milieu will likely differ from our *in vitro* setting, the amount of EVs per monocyte used here likely does not reflect the situation *in vivo*. Nonetheless, the use of amebic EVs as a model for immune responses during ALA allows for new experimental avenues in settings in which live parasites can't be easily used. This includes for example *in vitro* co-culture systems with immune cells and organoids that are complicated by different requirements of cell types in terms of culture media and oxygen exposure.

## Materials and methods

### Ethics statement

All mice used were bred in the animal facility of the Bernhard Nocht Institute for Tropical Medicine and kept in individually ventilated cages under pathogen-free conditions with a day/night cycle of 12 h, 21°C RT and 50 – 60% humidity. For harvesting of organs, C57BL/6J mice aged 9 – 13 weeks were euthanized by $CO_2$ overdose followed by cervical dislocation or cardiac puncture and blood withdrawal in accordance with German animal protection laws. Organ harvest was approved by the Animal Welfare Officer of the BNITM with acknowledgement of the Authority for Justice and Consumer Protection, Veterinary Affairs, Hamburg, Germany under the permission file numbers T-008 and T-011. Harvested organs were used for experiments immediately after euthanasia. Mice of both sexes were used for all experiments.

### Extracellular vesicles

*E. histolytica* trophozoites of clones A1 and B2 [22] were cultured axenically under microaerophilic conditions in TY-I-S-33 medium [92] supplemented with 1% 100x Penicillin/Streptomycin antibiotic mixture (Capricorn Scientific GmbH) at 37°C. For EV isolation, trophozoites were washed with sodium phosphate-buffered saline (NaPBS; 6.7 mM $NaHPO_4$, 3.3 mM $NaH_2PO_4$, 140 mM NaCl, pH 7.2) and resuspended in EV-depleted TY-I-S-33 medium, which was obtained

by ultracentrifugation at 100,000 $g$ for 18 h to ensure absence of serum-derived EVs. $2x10^5$ trophozoites/well were then cultured on collagen-coated 6-well plates (5 µg/cm$^2$ collagen from calf skin (Bornstein & Traub type I, Sigma-Aldrich C3511)) in 5 ml EV-depleted medium for 24 h (EV uptake experiments) or 46 h (EV isolation for other stimulation experiments) under anaerobic conditions (Anaerocult A, Merck KGaA) at 37°C. Medium without trophozoites was co-incubated as mock control. Subsequently, supernatants were harvested and subjected to differential centrifugation performed according to a protocol modified from Mantel *et al.* [93] (15 min centrifugation each at 600 $g$, 1600 $g$, 3600 $g$ and 10,000 $g$, supernatants processed for the next step and pellets discarded). EVs were pelleted in polypropylene tubes (open-top thinwall, Beckman Coulter 326823) by ultracentrifugation using Optima XE-90 centrifuge with SW 32 Ti swinging bucket rotor (Beckman Coulter) at 100,000 $g$ and 4°C for 1 h (maximal acceleration, deceleration 5) and washed once with PBS (136.9 mM NaCl, 2.7 mM KCl, 1.8 mM $KH_2PO_4$, 10 mM $Na_2HPO_4$, 0.22 µm filtered) with the same settings. EV pellets were finally resuspended in 200 µl PBS, aliquoted and stored at – 80°C. For mock control samples, all steps were performed the same way. For isolation of total RNA from EVs, 500 µl QIAzol reagent (QIAGEN) were added to 100 µl of EV sample before storage. For subsequent stimulation experiments, EVs of four separate EV isolations were combined into EV pools to minimize batch effects on stimulation. EV pools were again aliquoted before storage at −80 °C. EV pool aliquots were only thawed once for use in stimulation experiments and not re-frozen to minimize freeze-thaw cycles.

## Nanoparticle tracking analysis

Nanoparticle tracking was performed using Malvern Panalytical NanoSight LM14C with NTA 3.0 software according to the manual. EV samples were measured in a 1:100–1:300 dilution in 0.22 µm filtered PBS. A total of 900 frames were recorded over five separate 30 s measurements (camera level 16, 25 °C temperature control, detection threshold 6 for image processing).

## Immunogold labeling for transmission electron microscopy

EVs were spun down onto glow-discharged, carbon and formvar coated nickel grids (Plano GmbH) by centrifugation at full speed in a table-top centrifuge for 15 min, washed once with PBS and 4 times with 0.05% glycine in PBS for 3 min each. Blocking of unspecific binding was performed by incubation of the grids in blocking buffer (0.5% bovine serum albumin in PBS) for 10 min. Grids were subsequently incubated in primary antibody diluted in blocking buffer for 24 h at 4°C. Antibodies used were rabbit anti-Gal/GalNAc lectin (170 kDa subunit) [94,95] (1:200 dilution) and mouse anti-LPPG [96] (1:100 dilution). In order to control for antibody specificity, controls without primary antibody were performed (S1 Fig). After incubation, the grids were washed with blocking buffer 6 times for 3 min, followed by incubation in colloidal gold conjugated secondary antibody diluted 1:100 in blocking buffer for 24 h at 4°C (goat anti-mouse colloidal gold-conjugated secondary antibody 12 nm, Dianova GmbH, and goat anti-rabbit colloidal gold-conjugated secondary antibody 10 nm, Science Services GmbH). Grids were again washed with blocking buffer 4 times for 3 min, followed by 2 washes with $H_2O$ for 3 min. Sample fixation was performed using 2% glutaraldehyde for 5 min, followed by 2 washes with $H_2O$ for 3 min each. Finally, grids were incubated with 2% aqueous uranyl acetate for 15 s, washed once with $H_2O$ and dried at room temperature (RT). Sample imaging was performed using a Tecnai Spirit electron microscope at 80 kV.

## Mass spectrometry and proteome analysis

For trophozoite proteomes, *E. histolytica* were seeded on collagen-coated 6-well plates and incubated for EV isolation. Trophozoites were harvested by resuspension in cold NaPBS,

centrifuged at 400 *g* for 4 min, washed once with NaPBS and centrifuged as before, followed by storage of the pellet at – 80°C. EVs were processed for mass spectrometry (MS) as described above. The supernatant of the first ultracentrifugation step was used as negative control to determine proteins resulting from burst EVs or detached from the EV surface (S3 Table). Protein concentration of the samples was determined using Qubit 4 fluorometer (Thermo Fisher Scientific) according to the manual. 30 µg of protein for each sample were processed according to Hughes *et al.* [97] and Rappsilber *et al.* [98]. Tandem MS was performed with a Thermo Fisher Scientific Orbitrap Fusion (Q-OT-qIT) mass spectrometer. Raw MS data were processed using MaxQuant software (version 2.0.3.0) [99]. False discovery rate (FDR) was set to 1% for proteins and peptides and a minimum peptide length of seven amino acids was specified. Andromeda search engine in MaxQuant was used for spectra search against *E. histolytica* HM-1:IMSS annotated proteins (AmoebaDB release 56, https://amoebadb.org/amoeba/app [23]). The MaxQuant label-free algorithm was used for quantification [100]. Data annotation and statistical analysis of the MaxQuant output were performed with Perseus by MaxQuant [101]. Statistical comparison between two datasets was performed with Student's *t* test in Perseus with FDR *p*-value set to 0.05 and s0 = 0.5. Proteins present in only 1 out of 3 samples of a dataset were excluded from the proteome for downstream analysis. For comparison of EV proteomes with trophozoite proteomes, statistical overrepresentation test was performed with Panther knowledgebase 17.0 using Fisher's exact test with FDR-adjusted *p* value < 0.05 [27,28]. To identify amebic orthologs for the top 100 mammalian EV proteins from Vesiclepedia [26] (http://www.microvesicles.org/, version 5.1, accessed on 13 Aug 2024), human protein sequences were retrieved from UniProtKB (https://www.uniprot.org/) and orthologs identified through protein BLAST (blastp) against *E. histolytica*. For downstream analysis of proteomes and identification of transmembrane domains and signal peptides, AmoebaDB release 68 [23] (accessed on 16 Oct 2024) was used.

## miRNA sequencing and analysis

Total RNA was isolated from EV samples in QIAzol (QIAGEN) using miRNeasy Mini kit (QIAGEN) according to the manufacturer's instructions. RNA concentration was determined with RNA 6000 Pico Kit for 2100 Bioanalyzer (Agilent Technologies). Libraries for miRNA sequencing were constructed using NEXTFLEX small RNA-Seq Kit (PerkinElmer). Samples were sequenced at 27 – 132 million reads per sample with 50 bp read length using NovaSeq 6000 sequencing system with NovaSeq SP flow cell (Illumina). Raw data analysis and alignment to Zhang *et al.* [30] and Mar-Aguilar *et al.* [29] datasets was performed using CLC genomics workbench software version 24 (QIAGEN, https://digitalinsights.qiagen.com) using the Quantify miRNA tool version 1.3 with sequence length set to 15 – 40 nt or 15 – 30 nt, respectively. For this, Zhang *et al.* [30] reference data were accessed from NCBI gene expression omnibus (accession number GSE43668) and, for Mar-Aguilar *et al.* [29] dataset, a custom reference sequence list was created based on sequences listed in their supplementary information. *De novo* miRNA prediction from sequencing data was performed using BrumiR algorithm version 3.0 [31] with precursor clustering at 98% identity, followed by quantification and annotation using CLC genomics software. Out of the 3 samples of one clone, the largest dataset each was used for *de novo* miRNA prediction. miRNAs were considered true hits if they were present in at least 2 out of 3 samples with a minimum of 5 read counts. Target gene prediction was performed using miRanda algorithm (version 3.3a) [33,34] with *Homo sapiens* genome assembly GRCh38.p14 and *E. histolytica* HM-1:IMSS genome (AmoebaDB release 68 [23]) as reference. Interactions were filtered based on a pairing score of >150 and an energy score of ≤15 as described previously [102,103].

## Isolation of bone marrow-derived monocytes

Bones were sterilized by incubation in 70% isopropanol for 2 min, cut open and flushed with Dulbecco's PBS (DPBS, PAN-Biotech) using hypodermic needles to obtain bone marrow cells, which were separated using a 70 μm cell strainer. Monocytes were isolated from bone marrow cells using EasySep Mouse Monocyte Isolation Kit (StemCell Technologies) according to the manufacturer's instructions. Isolation efficacy was controlled by flow cytometry (S3A Fig). For this, isolated cells were stained with live/dead fixable blue dye (1:1000, Thermo Fisher Scientific) to determine viability and subsequently labeled with anti-CD11b (M1/70, Alexa Fluor 488-conjugated, 1:400, BD Biosciences 557672), anti-Ly6C (HK1.4, APC-conjugated, 1:200, BioLegend 128016) and anti-Ly6G (1A8, PE-conjugated, 1:400, BioLegend 127608) antibodies in Fc blocking solution. Samples were measured at Cytek Aurora spectral flow cytometer or BD Biosciences accuri C6 flow cytometer and analysis performed with FlowJo software version 10.

## Isolation of peripheral and bone marrow-derived neutrophils

For isolation of peripheral neutrophils, blood obtained by cardiac puncture was collected in EDTA-coated tubes and subjected to two rounds of erythrocyte lysis. Spleen cells were passed through a 70 μm cell strainer, washed with DPBS and subjected to one round of erythrocyte lysis. Immune cells from blood and spleen were then combined and passed through a 30 μm cell strainer prior to neutrophil isolation. Bone marrow cells were obtained as described for monocyte isolation above. Neutrophil isolation was performed using Neutrophil isolation kit (Miltenyi Biotec) according to the manufacturer's instructions. Isolation efficacy was controlled by flow cytometry using the same antibodies used for control of monocyte isolation (S3C Fig). Samples were measured at BD Biosciences accuri C6 flow cytometer and analysis performed with FlowJo software version 10.

## EV uptake

EVs were labeled during the EV isolation process after the first ultracentrifugation step with 10 μM 2',7'-Bis-(2-Carboxyethyl)-5-(and-6)-Carboxyfluorescein, Acetoxymethyl Ester (BCE-CF,AM) (Thermo Fisher Scientific) for 30 min, followed by PBS washing and ultracentrifugation. BCECF,AM is non-fluorescent until it is internalized in cells (or EVs) and cleaved by cytosolic esterases, yielding the fluorescent BCECF. Protein concentrations were determined using Qubit 4 fluorometer (Thermo Fisher Scientific) according to the manual. 0.5 μg of EVs or the corresponding mock control volume were added to $1x10^5$ bone marrow-derived monocytes in cRPMI (RPMI supplemented with 10% charcoal-stripped FCS, 1% penicillin/streptomycin, 1% L-glutamine) and incubated for 30 min. Stimulated monocytes were washed once with DPBS and fixed using eBioscience Foxp3/ Transcription Factor Staining Buffer Set (Thermo Fisher Scientific) according to the manufacturer's instructions. Samples were measured at Cytek Aurora spectral flow cytometer and analysis performed with FlowJo software version 10.

## EV stimulation

As quality control measure, EV pools were first tested for stimulatory capacity on crude bone marrow cells. $1 \times 10^6$ cells were stimulated with 1000 EVs/cell (concentration determined by NTA) or equal volume mock control in cRPMI for 24 h, followed by detection of IL-6 in culture supernatants using BD OptEIA Mouse IL-6 ELISA (BD Biosciences). Only EV pools eliciting increased concentrations of IL-6 upon stimulation compared with mock controls were

further used for experiments. $1 \times 10^5$ bone marrow-derived monocytes/peripheral neutrophils or $5 \times 10^5$ bone marrow-derived neutrophils were stimulated with EV pools at a concentration of 1000 EVs/cell in cRPMI for 8 h (for NGS) or 24 h (for flow cytometry analysis and cytokine assays). For heat inactivated (h.i.) controls, EV samples were incubated in a heating block for 10 min at 95 °C prior to use.

## Spectral flow cytometry for surface marker analysis

For the analysis of surface marker presence on 24 h stimulated monocytes using spectral flow cytometry, cells were washed with DPBS, stained with Zombie UV (1:1000, BioLegend) to determine viability and subsequently labeled with anti-CD11b (M1/70, BV510-conjugated, 1:400, BioLegend 101263), anti-Ly6C (HK1.4, PE-conjugated, 1:800, BioLegend 127608), anti-Ly6G (1A8, APC-conjugated, 1:400, BioLegend 127614), anti-CD38 (90, BV421-conjugated, 1:800, BioLegend 104445) and anti-CCR2 (SA203G11, PE-Cy7-conjugated, 1:100, BioLegend 150611) antibodies in Fc blocking solution. Labeled cells were then fixed using eBioscience Foxp3/ Transcription Factor Staining Buffer Set (Thermo Fisher Scientific) according to the manufacturer's instructions. Samples were measured at Cytek Aurora spectral flow cytometer and analysis performed with FlowJo software version 10.

## RNA-sequencing and data analysis

Monocytes stimulated with EVs for 8 h were fixed in RLT buffer and RNA was isolated using RNeasy Mini kit (QIAGEN) according to the manufacturer's instructions. RNA quality and concentration were assessed with RNA 6000 Pico kit for 2100 Bioanalyzer (Agilent Technologies). Only samples with a RNA integrity number (RIN) of >7.5 were processed for sequencing. Libraries were prepared using QIASeq Stranded mRNA Library kit (QIAGEN) and sequenced as 75 bp paired-end reads on Illumina NextSeq 550 system with NextSeq 500/550 Mid Output Kit v2.5 (150 cycles) (Illumina) at 5 – 6 million reads per sample. Raw data were mapped to *Mus musculus* reference genome GRCm39.112 and analyzed using CLC genomics Workbench software version 21 with RNA-Seq Analysis tool version 2.6 (QIAGEN). Genes that were not expressed in a minimum of 3 out of 4 samples with at least 20 read counts were excluded from downstream analyses. Volcano plots for the depiction of significantly differentially expressed genes were created with the European Galaxy Server (https://rna.usegalaxy.eu/) https://rna.usegalaxy.eu/ [104]. Heatmaps were created using Heatmapper (http://heatmapper.ca/) [105]. ShinyGO version 0.80 was used for GO term analysis (http://bioinformatics.sdstate.edu/go/) [37].

## Quantitative PCR

RNA was isolated from 8 h stimulated monocytes and freshly isolated monocytes (prestimulation) as described above and transcribed into cDNA using Maxima First Strand cDNA Synthesis kit (Thermo Fisher Scientific) according to the manual. Primers used for qPCR were *Ccl5*: forward 5-GGACTCTGAGACAGCACATG-3, reverse 5-GCAGTGAGGATGATGGTGAG-3, *Cxcl2*: forward 5-AGTTTGCCTTGACCCTGAAG-3, reverse 5-GGTCAGTTAGCCTTGCCTTT-3, *Oasl1*: forward 5-TGACGGTCAGTTTGTAGCCAT-3, reverse 5-AAATTCTCCTGCCTCAGGAAC-3, *Tnf*: forward 5-TCTGTGAAAACGGAGCTGAG-3, reverse 5-GGAGCAGAGGTTCAGTGATG-3, *Rps9*: forward 5-GCTAGACGAGAAGGATCCCC-3, reverse 5-TTGCGGACCCTAATGTGACG-3 (custom DNA oligos, Eurofins Genomics). Annealing temperature was determined by gradient PCR using Maxima SYBR Green/ROX qPCR Master Mix (Thermo Fisher Scientific) and LightCycler 96 (Roche) and set to 58°C. Primer pair efficiency was determined by qPCR on serial cDNA dilutions and calculated as

described in Pfaffl *et al.* [106]. Gene expression in EV-stimulated monocytes was quantified according to the Pfaffl method [106], with *Rps9* used as reference gene, and normalized to pre-stimulation control samples.

## Measurement of cytokine and MPO concentration

CCL3 and MPO concentrations were detected in immune cell supernatants after 24 h stimulation using Mouse CCL3/MIP-1 alpha DuoSet ELISA and Mouse Myeloperoxidase DuoSet ELISA kits (both R&D systems) according to the manufacturer's instructions. All other cytokines were assessed using Mouse M1 Macrophage LEGENDplex flow-based assay (CXCL1, IL-1β, IL-6, IL-12p40, IL-12p70, IL-18, IL-23, TNF; BioLegend) with BD LSRII flow cytometer (BD Biosciences). Only data for cytokines with concentrations above the lower limit of detection are shown.

## Statistics

All data statistically analyzed by GraphPad prism version 9 (this excludes sequencing data) were tested for normal distribution using Shapiro-Wilk test. Statistical significance between two groups was then tested using Student's *t* test in the case of NTA data. For datasets with multiple comparisons, statistical significance was tested using non-parametric Kruskal-Wallis test with Dunn's multiple comparisons test for data that did not pass the normality test or with parametric one-way ANOVA tests for normally distributed data. Brown-Forsythe test was used to test for statistically different standard deviations (SDs). Datasets with no significant differences in SDs were analyzed by ordinary one-way ANOVA with Šídák's multiple comparisons test, whereas datasets with significantly different SDs were analyzed with Brown-Forsythe and Welch's ANOVA with Dunnett's multiple comparisons test. Parametric tests are indicated as 'one-way ANOVA' in figure legends. Significance levels correspond to * $p < 0.05$, ** $p < 0.01$, *** $p < 0.001$, **** $p < 0.0001$.

## Supporting information

**S1 Fig.  Secondary antibody controls for immunogold labeling of *Eh*A1 and *Eh*B2 EVs.** (TIF)

**S2 Fig.  GO term analysis of cellular components enriched or depleted in EV compared with trophozoite proteomes.** Shown are selected GO terms associated with proteins enriched or depleted in EV proteomes compared with trophozoite proteomes, based on statistical over-representation test performed with Panther knowledgebase [27,28]. (TIF)

**S3 Fig.  Gating strategies for flow cytometry.** (A) Gating strategy used to control monocyte purity after isolation via flow cytometry. After gating for leukocytes, doublets were excluded by gating for FSC-A against FSC-H, followed by gating on live cells (live/dead blue-negative). Monocytes were identified as CD11b+Ly6C+Ly6G- cells and divided into cells expressing high amounts of Ly6C (Ly6C^hi) or low amounts of Ly6C (Ly6C^lo). Shown is a representative sample with a purity of 91.78% (CD11b+x(Ly6C^hi+Ly6C^lo)). (B) Gating strategy used on isolated monocytes to quantify uptake of BCECF-labeled EVs. (C) Gating strategy used to control neutrophil purity after isolation via flow cytometry. Neutrophils were identified as CD11b+Ly-6C+Ly6G+ cells. Shown is a representative sample of bone marrow-derived neutrophils with a purity of 98.4% (CD11b+xLy6G+). (TIF)

**S4 Fig. Flow cytometry analysis of marker expression on EV-stimulated monocytes.** Bone marrow-derived monocytes of male (dots in graphs) and female (triangles in graphs) mice were stimulated for 24h *in vitro* with 1000 EVs/cell or equal volume mock control and subsequently stained and analyzed by flow cytometry. To control for the effect of protein denaturation on stimulatory capacity, EV and control samples were heat inactivated (h.i.) at 95°C for 10min prior to stimulation. (A) Gating strategy used to identify marker expression. After gating for leukocytes, doublets were excluded by gating for FSC-A against FSC-H, followed by gating on live cells (Zombie UV-negative). Monocytes were identified as CD11b$^+$Ly6C$^+$Ly6G$^-$ cells and divided into cells expressing high amounts of Ly6C (Ly6C$^{hi}$) or low amounts of Ly6C (Ly6C$^{lo}$). Expression of activation marker CD38 and chemokine receptor CCR2 was determined on both Ly6C$^{hi}$ and Ly6C$^{lo}$ monocytes. Gates were set according to fluorescence minus one controls, shown here for CD38 and CCR2. (B) Percent live cells following EV stimulation based on the Zombie UV versus FSC-A gate in (A). (C, D) Percent Ly6C$^{hi}$ (C) and Ly6C$^{lo}$ (D) monocytes out of CD11b$^+$ cells. (E, F) Percent CCR2$^+$ Ly6C$^{hi}$ monocytes (E) and median fluorescence intensity (MFI) of Ly6C$^{hi}$CCR2$^+$ cells (F) following EV stimulation. (G, H) Percent CCR2$^+$ Ly6C$^{lo}$ monocytes (G) and median fluorescence intensity (MFI) of Ly6C$^{lo}$CCR2$^+$ cells (H) following EV stimulation. (Kruskal-Wallis test with Dunn's multiple comparisons test, $*p < 0.05$, n = 6). (TIF)

**S5 Fig. Pathways and interferon-stimulated genes affected by EV stimulation of monocytes.** Monocytes isolated from bone marrow of male and female mice were stimulated for 8h *in vitro* with 1000 EVs/cell or equal volume mock control. mRNA expression levels were subsequently analyzed using whole transcriptome sequencing. (A, B) KEGG pathway analysis of significantly upregulated genes in *Eh*A1 EV (A) of *Eh*B2 EV (B)-stimulated monocytes compared with mock controls (analysis performed with shinyGO version 80 [37], shown are the top 20 pathways). (C, D, E, F, G) Reads per kilobase per million mapped reads (RPKM) normalized expression values of selected interferon-stimulated genes in male- (dots) and female-derived (triangles) monocytes after EV stimulation compared with mock controls. (Bonferroni-corrected $p$ value, $*p < 0.05$, $*** p < 0.001$, $**** p < 0.0001$, n = 4). (TIF)

**S6 Fig. Quantification of cytokine and myeloperoxidase release upon EV stimulation.** Monocytes were stimulated for 24h *in vitro* with 1000 EVs/cell or equal volume mock control. Supernatants of stimulated cells were analyzed by ELISA or flow cytometry-based multiplex cytokine assay (LEGENDplex). To control for the effect of protein denaturation on stimulatory capacity, EV and control samples were heat inactivated (h.i.) at 95°C for 10min prior to stimulation. (A–C, G, and H) Median fluorescence intensities (MFIs) for cytokines as determined by LEGENDplex and (D–F, J, and K) resulting calculated concentrations. (I) CCL3 concentration in supernatants as determined by ELISA. (One-way ANOVA with Dunnett's or Šídák's multiple comparisons test and Kruskal-Wallis test with Dunn's multiple comparisons test, $* p < 0.05$, $** p < 0.01$, n = 3–6 (A–K)). (TIF)

**S1 Table. Comparison of *Eh*A1 and *Eh*B2 EV proteomes.** Marked in red are proteins present in both proteomes in significantly different amounts. Proteins highlighted in green are uniquely present in EV proteomes and were not detected in corresponding trophozoite proteomes. (XLSX)

**S2 Table. Molecular function GO term enrichment of proteins present in *Eh*A1 but not *Eh*B2 EVs.** GO term enrichment was performed with AmoebaDB release 60 [23]. (XLSX)

**S3 Table. Proteins present in negative controls.** Shown are proteins detected in a minimum of 2 out of 3 negative control samples.
(XLSX)

**S4 Table. Comparison of *Eh*A1 and *Eh*B2 trophozoite proteomes.** Marked in red are proteins present in both proteomes in significantly different amounts.
(XLSX)

**S5 Table. Proteins involved in EV biogenesis present in *Eh*A1 and *Eh*B2 EV proteomes.** ESCRT proteins and tetraspanins were identified according to López-Reyes et al. [24] and Tomii et al. [25], respectively.
(XLSX)

**S6 Table. Comparison of *Eh*A1 and *Eh*B2 EV proteomes to the Top 100 mammalian EV proteins.** Top 100 most commonly described EV proteins were accessed from Vesiclepedia [26] (http://www.microvesicles.org/, version 5.1). Human protein sequences were retrieved from UniProtKB (https://www.uniprot.org/) and orthologs identified through protein BLAST (blastp) against *E. histolytica*.
(XLSX)

**S7 Table. Analysis of GO terms enriched or depleted in EV proteomes compared with trophozoite proteomes.** Results according to statistical overrepresentation test in Panther knowledgebase version 17.0 [27,28].
(XLSX)

**S8 Table. Quantification of small RNA presence in *Eh*A1 and *Eh*B2 EVs.** miRNA sequencing data were annotated according to Zhang et al. [30] reference.
(XLSX)

**S9 Table. Differential expression analysis of novel miRNAs identified in *Eh*A1 and *Eh*B2 EVs.** Novel miRNAs were predicted in miRNA sequencing data using BrumiR algorithm version 3.0 [31].
(XLSX)

**S10 Table. Target prediction of novel *E. histolytica* miRNAs in the human and *E. histolytica* genome.** Targets were predicted using miRanda algorithm version 3.3a [33,34]. Shown are targets for selected miRNAs of interest (top 10 most abundant miRNAs present in EVs and Ehi-miR-200).
(XLSX)

**S11 Table. Bulk RNA sequencing results of EV-stimulated and control monocytes.** For each condition, samples 1 and 2 are monocytes from male mice, whereas samples 3 and 4 are monocytes from female mice.
(XLSX)

## Acknowledgments

We thank the staff of the BNITM animal facility as well as Heike Baum and Dr. Dániel Cadar of the BNITM sequencing facility for their help. We also thank Dr. Mohsin Shafiq and Dr. Andreu Matamoros-Angles of the Institute of Neuropathology, University Medical Center Hamburg Eppendorf for the use of their NanoSight as well as expert support with NTA. Thanks also go to current and former members of the research groups Molecular Infection Immunology and Host-Parasite Interaction for laboratory support and scientific advice.

## Author contributions

**Conceptualization:** Barbara Honecker, Hanna Lotter, Iris Bruchhaus.

**Formal analysis:** Barbara Honecker, Balázs Horváth, Karel Harant, Nahla Galal Metwally, Annika Bea, Stephan Lorenzen.

**Funding acquisition:** Hanna Lotter, Iris Bruchhaus.

**Investigation:** Barbara Honecker, Valentin A. Bärreiter, Katharina Höhn, Claudia Marggraff, Sören Franzenburg.

**Methodology:** Barbara Honecker, Stephanie Leyk.

**Supervision:** Hanna Lotter, Iris Bruchhaus.

**Visualization:** Barbara Honecker, Stephan Lorenzen.

**Writing – original draft:** Barbara Honecker.

**Writing – review & editing:** Barbara Honecker, Valentin A. Bärreiter, Juliett Anders, Stephanie Leyk, Maria del Pilar Martínez-Tauler, Annika Bea, Charlotte Hansen, Helena Fehling, Melanie Lütkemeyer, Hanna Lotter, Iris Bruchhaus.

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
