## [Decision Letter · Decision Letter 0]

31 Jan 2025

PNTD-D-25-00034*Entamoeba histolytica* extracellular vesicles drive pro-inflammatory monocyte signalingPLOS Neglected Tropical DiseasesDear Dr. Honecker, Thank you for submitting your manuscript to PLOS Neglected Tropical Diseases. After careful consideration, we feel that it has merit but does not fully meet PLOS Neglected Tropical Diseases's publication criteria as it currently stands. Therefore, we invite you to submit a revised version of the manuscript that addresses the points raised during the review process. In particular, two reviewers noted a need for re-writing and re-organizing the manuscript to improve readability and focus. Please submit your revised manuscript within 30 days Apr 01 2025 11:59PM. If you will need more time than this to complete your revisions, please reply to this message or contact the journal office at plosntds@plos.org. Please include the following items when submitting your revised manuscript: * A rebuttal letter that responds to each point raised by the editor and reviewer(s). You should upload this letter as a separate file labeled 'Response to Reviewers '. This file does not need to include responses to any formatting updates and technical items listed in the 'Journal Requirements' section below. * A marked-up copy of your manuscript that highlights changes made to the original version. You should upload this as a separate file labeled 'Revised Manuscript with Track Changes '. * An unmarked version of your revised paper without tracked changes. You should upload this as a separate file labeled 'Manuscript '. If you would like to make changes to your financial disclosure, competing interests statement, or data availability statement, please make these updates within the submission form at the time of resubmission. Guidelines for resubmitting your figure files are available below the reviewer comments at the end of this letter. We look forward to receiving your revised manuscript. Kind regards, Anjan Debnath, Ph.D.Guest EditorPLOS Neglected Tropical Diseases Laura-Isobel McCallSection EditorPLOS Neglected Tropical Diseases

Shaden Kamhawi

co-Editor-in-Chief

Paul Brindley

co-Editor-in-Chief

**Additional Editor Comments:** Thank you for your manuscript submission to PLOS Neglected Tropical Diseases. Below you will find the comments from the reviewers. The reviewers are overall satisfied with the work and they recognized the importance of this work to understand how E. histolytica communicates with host immune cells. The reviewers have raised some minor concerns and provided excellent suggestions to improve the quality of the manuscript. I hope you can address those concerns in your revised manuscript. **Journal Requirements:**

2) Some material included in your submission may be copyrighted. According to PLOSu2019s copyright policy, authors who use figures or other material (e.g., graphics, clipart, maps) from another author or copyright holder must demonstrate or obtain permission to publish this material under the Creative Commons Attribution 4.0 International (CC BY 4.0) License used by PLOS journals. Please closely review the details of PLOSu2019s copyright requirements here: PLOS Licenses and Copyright. If you need to request permissions from a copyright holder, you may use PLOS's Copyright Content Permission form.

Potential Copyright Issues:

- Figures 1A and 4A; Please confirm whether you drew the images / clip-art within the figure panels by hand. If you did not draw the images, please provide a link to the source of the images or icons and their license / terms of use; or written permission from the copyright holder to publish the images or icons under our CC BY 4.0 license. Alternatively, you may replace the images with open source alternatives. See these open source resources you may use to replace images / clip-art:

**Reviewers' comments:** Reviewer's Responses to Questions

**Key Review Criteria Required for Acceptance?**

**Methods**

-Are the objectives of the study clearly articulated with a clear testable hypothesis stated?

-Is the study design appropriate to address the stated objectives?

-Is the population clearly described and appropriate for the hypothesis being tested?

-Is the sample size sufficient to ensure adequate power to address the hypothesis being tested?

-Were correct statistical analysis used to support conclusions?

-Are there concerns about ethical or regulatory requirements being met?

Reviewer #1: The study aims to investigate how E. histolytica extracellular vesicles (EVs) influence pro-inflammatory signaling in monocytes, focusing on sex differences and their role in amebic liver abscess (ALA) pathology. The experimental design, involving a comparative analysis of EVs from low pathogenic (EhA1) and high pathogenic (EhB2) strains, aligns well with the study's objectives. Incorporating proteomic analysis, miRNA profiling, and cytokine assays further supports the research goals. The use of murine bone marrow-derived monocytes and neutrophils is an appropriate model, and the sample sizes for proteomics, RNA sequencing, and cytokine assays are sufficient to yield meaningful results. Statistical analyses, including ANOVA and multiple comparison corrections, have been appropriately applied, and ethical considerations for animal use have been addressed.

Reviewer #2: The goal here is to determine the role of exosomes released by Entamoeba trophozoites on monocytes, which are important component in liver infection in a mouse model. The two hooks they have are Entamoeba strains that cause little disease (EhA1) or severe disease (EhB2) and monocytes from female mice, which have modest liver disease (like female humans) or from male mice, which have severe liver disease (like male humans). While there is no genetic manipulation of Entamoeba or mice, there is extensive characterization of proteins and miRNAs of exosomes, as well as pro-inflammatory immune response of monocytes and myeloperoxidase (MPO) response of monocytes and neutrophils.

Reviewer #3: -Are the objectives of the study clearly articulated with a clear testable hypothesis stated?

Answer: Yes, in the part corresponding to the male and female monocytes stimulation with EVs.

-Is the study design appropriate to address the stated objectives?

Answer: Yes, in the part corresponding to the male and female monocytes stimulation with EVs.

-Is the population clearly described and appropriate for the hypothesis being tested?

Answer: Yes.

-Is the sample size sufficient to ensure adequate power to address the hypothesis being tested?

Answer: Yes

-Were correct statistical analysis used to support conclusions?

Answer: Yes

-Are there concerns about ethical or regulatory requirements being met?

Answer: NO

**Results**

-Does the analysis presented match the analysis plan?

-Are the results clearly and completely presented?

-Are the figures (Tables, Images) of sufficient quality for clarity?

Reviewer #1: The results presented in this article make an important contribution to understanding how the E. histolytica parasite communicates with the host's immune cells. A comprehensive omics approach has been used to characterize the content of E. histolytica EVs and their effect on monocyte activation markers. While the results are clearly presented, some sections are dense; for instance, the proteomic comparisons could benefit from a more concise summary. The figures are of high quality and effectively illustrate the key findings.

Reviewer #2: Their proteomic data shows that exosomes of the two Eh strains have expected proteins, as well as unique proteins. A strength here is that the exosome proteins are not contaminated by broken cells, and there are many differences between the strains. A weakness here is that there are so many hypothetical proteins, which they should at least attempt to identify using AlphaFold and Foldseek. It is not clear how useful the miRNA data from exosomes, as there are so many targets in Eh and in host cells. Transcript data from monocytes exposed to exosomes of virulent and avirulent Eh show expected increases in cytokines, which is greater in monocytes of male mice that have more severe liver disease. Further, the major finding of the paper, which was not expected, is that exosomes of avirulent EhA1 cause increased release of MPO in monocytes and neutrophils, which plays a protective role in Eh infection.

Reviewer #3: -Does the analysis presented match the analysis plan?

Answer: No, because authors’ aim is to study the role of EVs on male and female monocytes during the hepatic abscess formation due to E. histolytica infection and an important part of the paper is focused to define the EV’s content.

-Are the results clearly and completely presented?

Answer: No, the part corresponding to the EVs content is confuse and the link of the proteins and iRNA finding with monocytes activation is not clear.

-Are the figures (Tables, Images) of sufficient quality for clarity?

Answer: Yes

**Conclusions**

-Are the conclusions supported by the data presented?

-Are the limitations of analysis clearly described?

-Do the authors discuss how these data can be helpful to advance our understanding of the topic under study?

-Is public health relevance addressed?

Reviewer #1: The conclusions are consistent with the presented data. However, while the discussion touches on experimental limitations, it does not address the impact of EV isolation methods on the results (see the following article as an example: https://doi.org/10.1016/j.jcyt.2023.11.001). Including a discussion on this matter would be a valuable addition to the manuscript.

Reviewer #2: Yes, their conclusion that exosomes secreted by virulent Eh2B strain activate more cytokines in monocytes from male mice is supported by their data. Further they make the unexpected discovery that MPO is released by monocytes activated by exosomes of the less virulent Eh1A strain. In their lengthy discussion, they do not talk enough abut the limitations of their results (e.g., the absence of tight links between components of exosomes and responses of monocytes). Nor do they concede the absence of experimental manipulation of either amoebae or the host.

Reviewer #3: -Are the conclusions supported by the data presented?

Answer: Yes

-Are the limitations of analysis clearly described?

Answer: No

-Do the authors discuss how these data can be helpful to advance our understanding of the topic under study?

Answer: Yes and without any doubt the part corresponding to the monocytes activation is interesting and relevant for the field, but the first part of the Results section is too long and the enormous number of molecules identified make lose the focus of the main aims.

-Is public health relevance addressed?

Answer: Yes, amoebiasis continues to be a health problem in Mexico, India and other countries. It is necessary to continue studying the factors that cause its virulence to find a way to defeat it.

**Editorial and Data Presentation Modifications?**

Reviewer #1: I strongly recommend summarizing the dense data sections, particularly in the proteomics part and the discussion, to improve readability.

Reviewer #2: Is not clear whether they have deposited proteomic data to any accessible database. Further, it would be best to present peptide coverage for proteins in an Excel file, so one could judge relative abundance of proteins in exosomes.

Reviewer #3: Divide the paper in two. Leave in this all the related to the monocyte activation by EVs and the EVs characterization publish elsewhere.

**Summary and General Comments**

Reviewer #1: The study is a well-conducted investigation into the role of E. histolytica EVs in immune modulation. Its strengths lie in its robust methodology and relevance to neglected tropical diseases. However, the manuscript could benefit from improving the readability of the manuscript particularly in the proteomics part and the discussion.

Reviewer #2: This reviewer is not asking for new experiments, because an awful lot of work has already been done and three major results are clear. First, exosomes are different for avirulent versus virulent strains of Eh. Second, cytokine activation is also different for monocytes from male mice, which has worse disease, than from female mice. Third, exosomes from avirulent Eh induce an MPO response in monocytes that likely reduce liver pathology. Yes, the links between precise exosomal proteins and monocyte response are absent, but it is a good start.

Reviewer #3: In this paper authors propose to study the interaction of E. histolytica trophozoites with murine monocytes and its effect in hepatic abscesses formation. They also assessed myeloperoxidase released in neutrophils by extracellular vesicles stimulation. Authors have used two E. histolytica populations with different virulence to compare the effect and content of the EVs that these trophozoites secrete. The statement of the hypothesis and the strategy used are well supported and are adequate to resolve the scientific questions that the authors pose. Authors use to perform their study two E. histolytica clones with different virulence, their experiments are well performed with the adequate controls and correct statistical analysis of the results.

There are studies supporting the fact that immune response in amoebiasis, mainly in liver infection has been documented by several authors, including Zhen et al and others. Thus, this manuscript will add new insights to the knowledge of the role of the immune response in the parasitic infection and it is relevant for public health.

However, the manuscript addresses two distinct issues: n EVs content and role of male and female murine monocytes in infection. The section corresponding to EVs results cumbersome with many not sufficiently explained data. I did not find a clear correlation between the enormous number of proteins and iRNA detected in vesicles with the main objective of the paper that is to study the monocytes role in amoebiasis. Which protein or proteins stimulate monocytes? What means the differences between EVs from pathogenic and non-pathogenic trophozoites?

In addition, to deep in the study ofn EVs, authors should use the exosomes markers described for E. histolytica and other eukaryotic cells, such as Vps23.,

PLOS authors have the option to publish the peer review history of their article (what does this mean? ). If published, this will include your full peer review and any attached files.

**Do you want your identity to be public for this peer review?** For information about this choice, including consent withdrawal, please see our Privacy Policy .

Reviewer #1: No

Reviewer #2: No

Reviewer #3: No

---

## [Decision Letter · Decision Letter 1]

19 Mar 2025

Dear Ms Honecker,

We are pleased to inform you that your manuscript '*Entamoeba histolytica* extracellular vesicles drive pro-inflammatory monocyte signaling' has been provisionally accepted for publication in PLOS Neglected Tropical Diseases.

Best regards,

Anjan Debnath, Ph.D.

Guest Editor

Laura-Isobel McCall

Section Editor

Shaden Kamhawi

co-Editor-in-Chief

Paul Brindley

co-Editor-in-Chief

Reviewer's Responses to Questions

**Key Review Criteria Required for Acceptance?**

**Methods**

-Are the objectives of the study clearly articulated with a clear testable hypothesis stated?

-Is the study design appropriate to address the stated objectives?

-Is the population clearly described and appropriate for the hypothesis being tested?

-Is the sample size sufficient to ensure adequate power to address the hypothesis being tested?

-Were correct statistical analysis used to support conclusions?

-Are there concerns about ethical or regulatory requirements being met?

Reviewer #1: (No Response)

Reviewer #2: (No Response)

**Results**

-Does the analysis presented match the analysis plan?

-Are the results clearly and completely presented?

-Are the figures (Tables, Images) of sufficient quality for clarity?

Reviewer #1: (No Response)

Reviewer #2: (No Response)

**Conclusions**

-Are the conclusions supported by the data presented?

-Are the limitations of analysis clearly described?

-Do the authors discuss how these data can be helpful to advance our understanding of the topic under study?

-Is public health relevance addressed?

Reviewer #1: (No Response)

Reviewer #2: (No Response)

**Editorial and Data Presentation Modifications?**

Reviewer #1: (No Response)

Reviewer #2: (No Response)

**Summary and General Comments**

Reviewer #1: (No Response)

Reviewer #2: (No Response)

PLOS authors have the option to publish the peer review history of their article (what does this mean? ). If published, this will include your full peer review and any attached files.

**Do you want your identity to be public for this peer review?** For information about this choice, including consent withdrawal, please see our Privacy Policy .

Reviewer #1: No

Reviewer #2: **Yes: ** John Samuelson

---

## [Editor Report · Acceptance letter]

Dear Ms Honecker,

We are delighted to inform you that your manuscript, "*Entamoeba histolytica* extracellular vesicles drive pro-inflammatory monocyte signaling," has been formally accepted for publication in PLOS Neglected Tropical Diseases.

Best regards,

Shaden Kamhawi

co-Editor-in-Chief

Paul Brindley

co-Editor-in-Chief
